# Statistical laws of stick-slip friction at mesoscale

Caishan Yan [1], Hsuan-Yi Chen [2,3], Pik-Yin Lai [2,3] & Penger Tong [1] ✉

Friction between two rough solid surfaces often involves local stick-slip events occurring at different locations of the contact interface. If the apparent contact area is large, multiple local slips may take place simultaneously and the total frictional force is a sum of the pinning forces imposed by many asperities on the interface. Here, we report a systematic study of stick-slip friction over a mesoscale contact area using a hanging-beam lateral atomic-force-microscope, which is capable of resolving frictional force fluctuations generated by individual slip events and measuring their statistical properties at the single-slip resolution. The measured probability density functions (PDFs) of the slip length $\delta x_s$, the maximal force $F_c$ needed to trigger the local slips, and the local force gradient $k'$ of the asperity-induced pinning force field provide a comprehensive statistical description of stick-slip friction that is often associated with the avalanche dynamics at a critical state. In particular, the measured PDF of $\delta x_s$ obeys a power law distribution and the power-law exponent is explained by a new theoretical model for the under-damped spring-block motion under a Brownian-correlated pinning force field. This model provides a long-sought physical mechanism for the avalanche dynamics in stick-slip friction at mesoscale.

Stick-slip is a common phenomenon both in nature and in many engineering applications. It is often observed in out-of-equilibrium disordered systems as a yield response to a smoothly varying external force and is characterized by intermittent bursts of irregular signals of different amplitudes, durations, and separations that result from the spontaneous depinning of mechanical contacts or local rearrangement of material bonds[1–4]. Among the stick-slip phenomena that have been explored thus far, friction between two (rough) solid surfaces in contact has attracted special attention[5–9] partly because of its connection with earthquakes, whose impact on our living conditions is often catastrophic. Earthquake on faults is believed to be a result of frictional stick-slip instabilities along pre-existing faults[10–13].

A common feature of stick-slip events is their broad range of slip sizes, manifest as power-law distribution of many orders of magnitude[4,12,13]. Observations that many different systems behave in a similar manner have prompted many theoretical, numerical, and experimental investigations aimed at finding some common mechanism or universal law underpinning these phenomena, which are also referred to as avalanche dynamics[4,12–15]. Because of the complexity of surface topologies and complex material parameters involved, our fundamental understanding of stick-slip friction is often limited by fewer direct measurements of individual slip events and oversimplified modeling and simulations conducted thus far[12,13,16–19]. It remains unclear whether stick-slip friction is indeed operating at a critical point, and if so, how the avalanche scaling is determined by the basic features of the surface roughness landscape?

Experimental studies of the friction law of solid interfaces, such as those reported in refs. 5,6,11,20–22 involved a macroscopic solid surface sliding against a rough substrate. Because the contact area of the interface contains a large number $n$ of random asperities, the measured frictional force fluctuations resulting from individual slips were usually averaged out as $1/\sqrt{n}$. Consequently, these studies focused

[1]Department of Physics, Hong Kong University of Science and Technology, Clear Water Bay, Kowloon, Hong Kong. [2]Department of Physics and Center for Complex Systems, National Central University, Taoyuan City 320, Taiwan. [3]Physics Division, National Center for Theoretical Sciences, Taipei 10617, Taiwan. ✉e-mail: penger@ust.hk

mainly on the macroscopic laws of friction, which involve aging of the static friction coefficient and the velocity-dependence of the dynamic friction coefficient[5,6,21,22]. Another type of experiment used an atomic force microscope (AFM) to measure atomic-scale friction with a nanoscopically sharp AFM tip sliding against a single-crystal surface[23–27]. Because the contact area between the AFM tip and the substrate is so small, only a single asperity is involved for a given slip event. These AFM experiments thus studied single-asperity dynamics without involving any collective effect of multi-asperity slips, which is a key factor in understanding the collective stick-slip dynamics[6,12,13].

In this work, we demonstrate that the hanging-beam AFM shown in Fig. 1 below provides a versatile experimental framework at the mesoscale that is small enough to resolve individual slip events but is also large enough to examine a broad range of slip sizes in a well-characterized disorder landscape with a single-slip resolution. With this framework, we are able to resolve frictional force fluctuations generated by individual slip events and provide a statistical description of stick-slip friction, which bridges the gap between the microscopic behavior of individual slips and the macroscopic laws of friction for solid interfaces.

## Results

Figure 1a shows the working principle of the long-beam AFM. A vertical hanging beam made with a thin rectangular cantilever beam is glued onto the front end of a horizontal AFM cantilever with the normal direction of the beam plane in parallel with the AFM scanning direction (x-direction). The free end of the hanging beam is coated with a drop of UV-cured glue mixed with 16 wt% glass nanoparticles to increase the compliance of the contact and at the same time, reduce adhesion to the substrate. The end surface of the scanning probe in contact with the substrate is milled by using a focused ion beam (FIB) so that it has two specific shapes and dimensions: one has a line shape with dimensions $34 \times 3\,\mu m^2$ ("quasi-1D probe" with an aspect ratio 11:1, Fig. 1b) and the other has a square shape with dimensions $12 \times 12\,\mu m^2$ ("2D probe", Fig. 1c) (see "Methods" below for more details).

The substrate used in the experiment is an ultra-fine silicon carbide sandpaper having an average grain size of 100 nm, which is taped onto a flat glass slide. Figure 1d shows an AFM topographical image of the sandpaper surface, from which we find its RMS roughness to be 99.7 nm over an area of $10 \times 10\,\mu m^2$ (see Supplementary Information (SI) Section I.A for more details). The surface of the silicon carbide sandpaper is hydrophobic, and its adhesion to the scanning probe is small compared with the applied normal load under ambient conditions (see SI Section II.B for more details). When the scanning probe is placed against the sandpaper under a normal load $N$ and slides laterally along the x-direction at a low-speed $U$, we measure the frictional force $F(x)$ between the two contact surfaces as a function of the scan

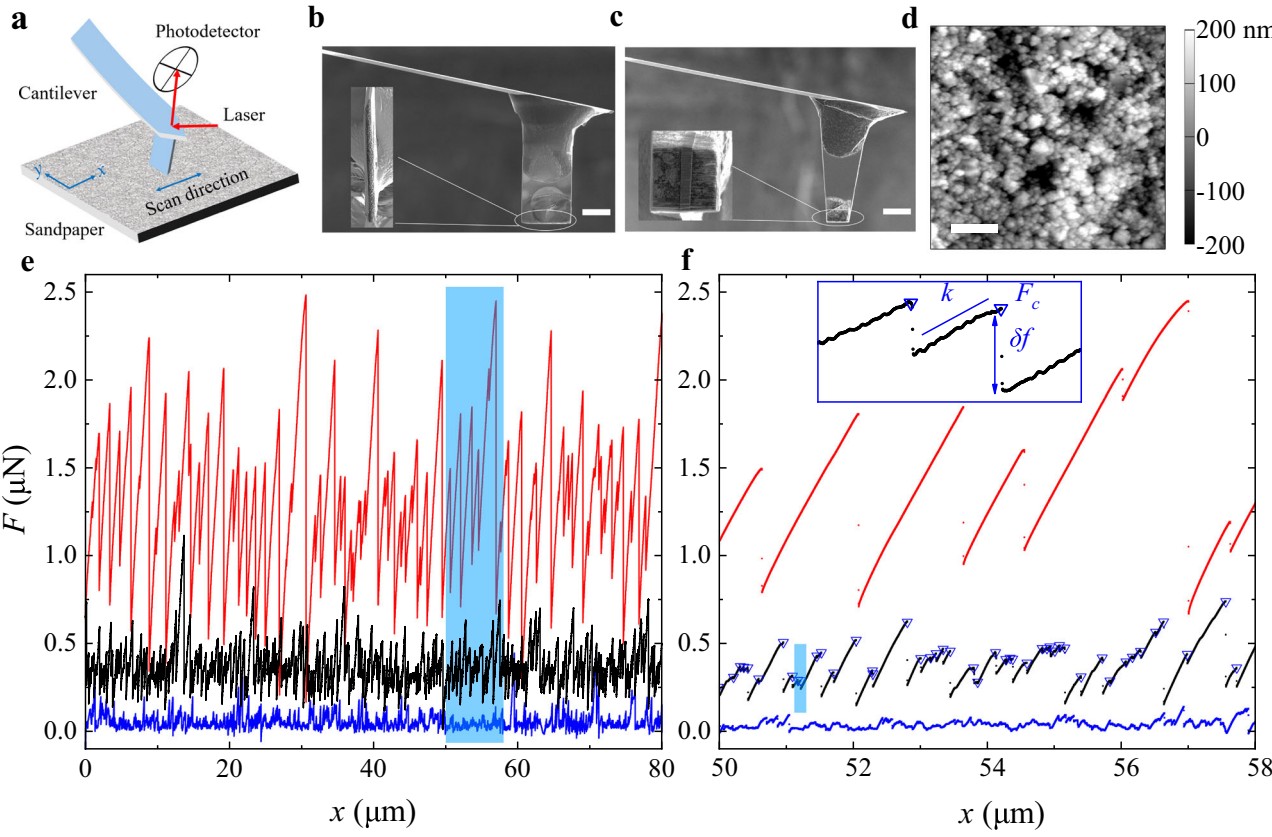

**Fig. 1 | Experimental setup and measurement of the frictional force curve. a** A sketch of the hanging-beam AFM for the measurement of the frictional force $F(x)$ between the end surface of the scanning probe and the substrate. **b** An SEM image showing the side and end (inset) views of a quasi-1D scanning probe with an end contact area of $(34 \pm 2) \times (3.0 \pm 0.5)\mu m^2$. **c** An SEM image showing the side and end (inset) views of a 2D scanning probe with an end contact area of $(12 \pm 1) \times (12 \pm 1)$ $\mu m^2$. The end-view SEM image reveals that the thin cantilever beam is embedded in the middle of the end portion of the scanning probe. **d** An AFM topographical image of the ultra-fine sandpaper surface ($10 \times 10\,\mu m^2$) with a nominal grain size of 100 nm. The vertical grayscale indicates the surface height. The scale bar in **b** and **c** is 20 μm, and that in **d** is 2 μm. **e** Measured force trajectories $F(x)$ as a function of

scan displacement $x$ under three different normal loads: $N = 100$ nN (blue curve), 500 nN (black curve), and 2400 nN (red curve). For clarity, the black and red curves are shifted upwards by 0.15 μN and 0.5 μN, respectively, which become their new zero-points in the vertical axis. The measurements are made using the 2D probe at the same scanning speed $U = 100$ nm/s. **f** A magnified view of the force trajectories in the blue-shaded region in (**e**). The blue downward triangles on the black curve mark the onset of each individual slip event. The inset shows two stick-slip events in the blue-shaded area of the black curve (at $x \simeq 51.1\,\mu m$). Here $k$, $F_c$, and $\delta f$ denote, respectively, the dynamic spring constant in the steady state, the maximum force needed to trigger a slip and force release during a slip. Source data are provided as a Source Data file.Source Data.

displacement $x = Ut$ of the probe [Fig. 1e, f and Supplementary Figs. 5–8 in SI Section I.D]. Because Young's modulus of the nanoparticles-embedded UV-cured glue (~3 MPa) is much smaller than that of the sandpaper substrate (see "Methods" below for more details), the contact between the end surface of the probe and the sandpaper is improved when the normal load $N$ is applied in the range studied.

Figure 1e shows how the measured force trajectory $F(x)$ changes with increasing normal load $N$. Because the scanning probe is under a steady-state motion over the sandpaper, the measured $F(x)$ fluctuates around a constant mean for a given $N$. The variations of $F(x)$ exhibit different characters, however, for different $N$, as shown in Fig. 1f. At a small load with $N = 100$ nN (blue curve), the measured $F(x)$ varies smoothly with displacement $x$ most of the time and large fluctuations are observed only occasionally. This behavior indicates that the scanning probe is under a continuous sliding, a regime which has been studied in previous experiments[5,8,27,28] (see SI Section II.G for more experimental results). At an intermediate load with $N = 500$ nN (black curve), $F(x)$ shows sawtooth-like fluctuations with a slow linear accumulation of force (stick) followed by a sharp force release (slip), which is characteristic of the stick-slip motion.

Prandtl and Tomlinson (PT) proposed[29–31] that the transition from smooth (or continuous) sliding to stick-slip takes place when $(2\pi^2 E_b)/(k_0 \lambda^2) > 1$, where $E_b$ and $\lambda$ are, respectively, the height and width of the potential trap associated with the surface roughness and $k_0$ is the effective (lateral) spring constant of the scanning probe. The maximal lateral force $F_c$ needed to overcome the energy barrier $E_b$ can be estimated as $F_c = \mu_c N_m \simeq \pi E_b / \lambda$[27,32], where $\mu_c$ is the friction coefficient associated with $F_c$. The PT model thus predicted $N_m > k_0 \lambda / (2\pi \mu_c)$. For our system, we have $k_0 \simeq 0.48$ N/m (see Fig. 3 below), the lower cutoff size of the grain clusters $\lambda \simeq 0.34$ μm (see SI Section I.A for more details), and $\mu_c \simeq 0.26$ (see SI Section II.C for more details), and hence $N_m > 100$ nN. This estimate is consistent with our observations.

At the high load limit with $N = 2400$ nN $\gg N_m$ (red curve), the stick-slip is predominated by a smaller number of large-amplitude slip events with a dramatic increase of the force drop $\delta f$ during each slip and a large reduction of the slip incident rate, as shown by the red curve in Fig. 1f. Similar effects were also observed in other AFM[28] and macroscopic[5] stick-slip experiments. Under a large normal load, the contact geometry at the solid interface may change significantly because nearby individual micro-contact areas start to interact, and some of them may coalesce and become large ones[33,34]. As a result, the scanning probe senses fewer but larger asperities, and the incidence rate of slips is reduced compared to that at an intermediate $N$ (see SI Section II.E for more details).

In this study, we focus on the measurements in the intermediate range of the normal load $N = 200$–$600$ nN, in which the scanning probe remains at an "optimal contact" with the sandpaper so that it can sense the full range of the rough landscape quasi-statically with negligible wear (see SI Section I.D and Section II.E for more details). Individual stick-slip events in this regime can be characterized by three quantities: $k$, $F_c$, and $\delta f$, as marked in the inset of Fig. 1f (see SI Section II.A for more details). The slope $k$ of the linear force accumulation defines the dynamic spring constant of the scanning probe. When $F(x)$ reaches its local maximal value $F_c$, some of the pinning sites in the contact area can no longer be held by the asperities, such that they slip off with a sudden release of the elastic restoring force $\delta f$. In this way, the local contact region moves forward, and this process repeats itself when the contact region becomes pinned again at a new position. Because the pinning force field associated with the sandpaper is random, the three quantities, $k$, $F_c$, and $\delta f$, all exhibit significant fluctuations, as shown by the black curve in Fig. 1f. As will be shown below, at the optimal contact, the stick-slip motion is at a critical point[14,18], at which the measured $k$, $F_c$, and $\delta f$ reveal universal statistical properties.

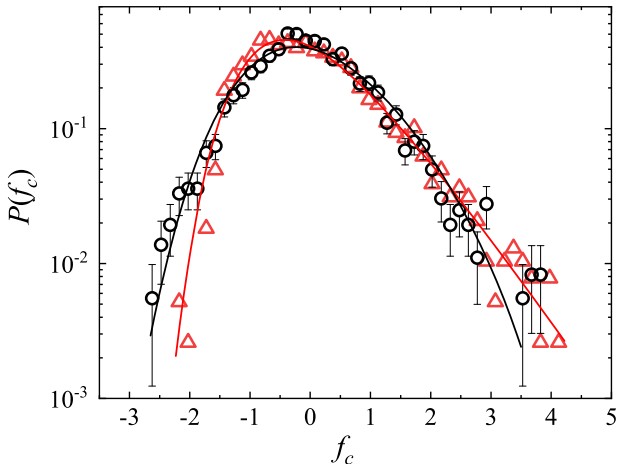

**Fig. 2 | Statistics of the maximal depinning force.** Measured probability density function (PDF) of the normalized maximal force $f_c$. The measurements are made at the same scanning speed $U = 100$ nm/s, and under the same normal load $N = 500$ nN, for both the quasi-1D probe (black circles) and 2D probe (red triangles). For each set of data, the measured $F(x)$ recorded ~2500 slip events over a traveling distance $x \simeq 500$ μm. The black and red solid lines show, respectively, the fits of Eq. (1) to the black circles with $\xi = -0.13 \pm 0.06$ and to the red triangles with $\xi = -0.03 \pm 0.05$. The error bars show the standard deviation of the black circles. Source data are provided as a Source Data file.Source Data.

Figure 2 shows the measured probability density function (PDF) $P(f_c)$ of the normalized maximal force $f_c = (F_c - \langle F_c \rangle)/\sigma_{F_c}$, where $\langle F_c \rangle$ and $\sigma_{F_c}$ are, respectively, the mean and standard deviation of $F_c$. It is seen that the measured PDFs $P(f_c)$ for the quasi-1D probe (black circles) and 2D probe (red triangles) overlap with each other, suggesting that the measured $P(f_c)$ is an intrinsic property of the stick-slip friction on the sandpaper and is not sensitive to the shape and dimensions of the scanning probe used. The two sets of data can be well described by the generalized extreme value (GEV) distribution[35],

$$P(f_c) = \frac{1}{\beta}(1 + \xi z)^{-(1+1/\xi)} e^{-(1+\xi z)^{-1/\xi}},  \quad (1)$$

where $z = (f_c - \mu)/\beta$ is the standardized variable with $\beta = |\xi|/\sqrt{\Gamma(1 - 2\xi) - \Gamma^2(1 - \xi)}$ and $\mu = \beta[1 - \Gamma(1 - \xi)]/\xi$ being, respectively, the scale and location parameters of the distribution function, and $\Gamma(x)$ is the gamma function. Equation (1) is used to model the distribution of extreme values in a sequence of independent and identically distributed random variables with zero mean and unity variance so that the normalized maximal force $f_c$ is used as the random variable in Eq. (1) (see SI Section II.D.1 for more details). The solid lines in Fig. 2 show the fits of Eq. (1) to the data points with only one fitting parameter $\xi$. For the data obtained with the quasi-1D probe (black circles), we find $\xi = -0.13 \pm 0.06$. For the data obtained with the 2D probe (red triangles), we find $\xi = -0.03 \pm 0.05$. For $\xi = 0$, Eq. (1) is reduced to the Gumbel distribution, which has an exponential tail at large values of $z$ without an upper bound for the local maximal force. Evidently, the data shown in Fig. 2 are not far from this limit. Figure 2 thus demonstrates that the maximal force $F_c$ is a depinning force for local slips, which follows the extreme value statistics.

When the scanning probe is pulled under a constant speed $U$, the interface motion is accomplished by a continuous series of local wiggling of individual (microscopic) contact areas over the asperities on the sandpaper via many individual stick-slip events of different sizes across the entire (macroscopic) contact area. A sudden slip of the contact interface takes place only locally for certain individual contact areas where the asperity-induced pinning force can no longer balance

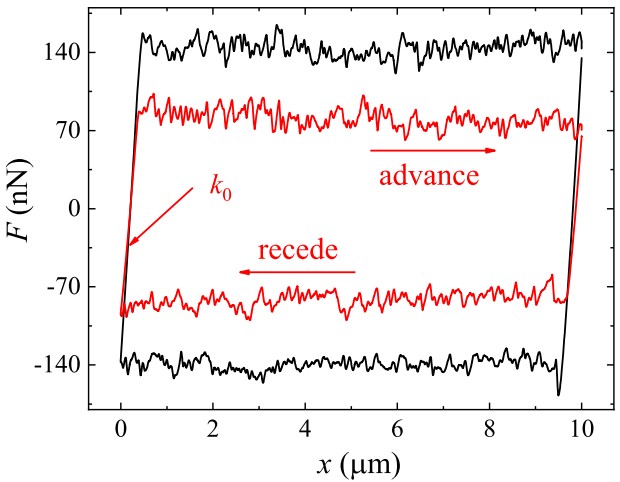

**Fig. 3 | Frictional force loops measured on a smooth silicon wafer surface.** The force loops are measured as a function of traveling distance $x$ when the quasi-1D probe (black curve) and 2D probe (red curve) are pulled to advance ( → ) and recede ( ← ) for a whole cycle against a smooth silicon wafer surface, which is coated with a thin layer of gold. The measurements are made under the same normal load $N = 200$ nN and at the same scanning speed $U = 1$ μm/s. In the plot, $k_0$ denotes the slope of the force curve on the left and right sides of the force loop. Source data are provided as a Source Data file. Source Data.

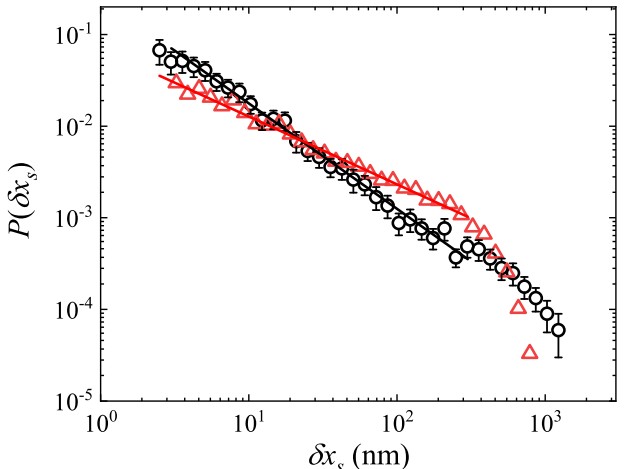

**Fig. 4 | Statistics of the slip length.** Log–log plot of the measured PDF of the interface displacement $\delta x_s$ associated with each slip. The measurements are made at the same scanning speed $U = 100$ nm/s, and under the same normal load $N = 500$ nN, for both the quasi-1D probe (black circles) and 2D probe (red triangles). The solid lines show the power-law fits of Eq. (2) to the black circles with the power-law exponent $\tau = 1.12 \pm 0.10$ and to the red triangles with $\tau = 0.72 \pm 0.10$. The error bars show the standard deviation of the black circles. Source data are provided as a Source Data file. Source Data.

the continuously increasing elastic restoring force resulting from the deformed solid interface (on the probe side). The local slip will be stopped eventually by some large asperities in front of the moving interface. As will be shown below (see the Discussion Section below for details), the measured force release $\delta f$ is related to the interface displacement $\delta x_s$ associated with each slip by $\delta f \simeq k_0 \delta x_s$, where $k_0$ is the static spring constant of the scanning probe, when it is completely pinned at the interface.

The value of $k_0$ can be obtained by measuring the frictional force loop when the scanning probe is pulled to move back and forth for a whole cycle against a smooth silicon wafer surface coated with a thin

layer of gold, as shown in Fig. 3. The gold-coated silicon wafer surface has an RMS roughness of 1.0 nm over an area of 10 μm × 10 μm (see SI Section I.A for more details). Before the probe starts to move, it is pushed downward against the wafer surface under a normal load $N = 200$ nN, and the contact between the two solid surfaces is tight. When the scanning probe advances ( → ) (or recedes ( ← )), it is stretched first, causing a sharp linear increase (or decrease) in $F$ with the distance $x$ traveled, as shown by the straight lines on the left (or right) side of the force loops. From the slope of the linear curve, we find the static spring constant $k_0 \simeq 0.66$ N/m for the quasi-1D probe (black curve) and $k_0 \simeq 0.48$ N/m for the 2D probe (red curve). When the pulling force reaches a critical value $F_c$, the interface de-pins and begins to move. Because the substrate is smooth, only smooth sliding with small force fluctuations, instead of stick-slip motion, is observed. This is shown by the horizontal plateau on the top (or bottom) of the force loops.

Figure 4 shows the measured PDF $P(\delta x_s)$ of the slip length $\delta x_s$ on the sandpaper, which is the normalized force release $\delta f/k_0$ during each slip. In contrary to Fig. 2, the measured $P(\delta x_s)$ for the quasi-1D probe (black circles) does not overlap with that for the 2D probe (red triangles), but both sets of data can be described by a power-law distribution,

$$P(\delta x_s) \sim (\delta x_s)^{-\tau}. \tag{2}$$

The value of the power-law exponent $\tau$ changes with the probe geometry and dimensions, and we find $\tau = 1.12 \pm 0.10$ for the quasi-1D probe and $\tau = 0.72 \pm 0.10$ for the 2D probe. When a mesoscale scanning probe slides against the sandpaper under a proper normal load, the contact area contains a finite number of asperities (say, 100–1000 asperities), and stick-slip friction involves multi-asperity slips. For example, when a strong asperity slips, it releases a large stress, which is partially transferred to its neighboring asperities and triggers their slips. This is an avalanche process, which gives rise to a broad range of slip lengths $\delta x_s$ without a characteristic value[4,12–15]. The power-law exponent $\tau$ quantifies the relative incidence between the large-sized and small-sized slips, i.e., more large-sized slips are detected for a smaller value of $\tau$, and vice versa. Figure 4 thus demonstrates that stick-slip friction indeed involves avalanche dynamics so that the slip lengths (or sizes) $\delta x_s$ obey a power-law distribution without a characteristic value (see SI Section II.D, Section II.E and Section II.F for more experimental results).

## Discussion

To explain the above experimental results, we now consider the force release $\delta f$ for a slip, which can be written as, $\delta f = \int_{A(x_s)} \sigma(x, y; x_s) dA - \int_{A(x_s + \delta x_s)} \sigma(x, y; x_s + \delta x_s) dA$, where $\sigma(x, y; x_s)$ is the asperity-induced heterogeneous interfacial shear stress at the contact surface of area $A = L_x L_y$ and Cartesian coordinates $(x, y)$, and $x_s$ is the center-of-mass position of the contact surface with $\delta x_s$ being its displacement during the slip. By introducing an effective hetero-geneous interfacial tension $h(y; x_s) = \int_{L_x} dx \sigma(x, y; x_s)$, we have

$$\delta f = \int_{L_y} dy [h(y; x_s) - h(y; x_s + \delta x_s)]$$
$$\simeq -\langle \partial_{x_s} h \rangle L_y \delta x_s \simeq k_0 \delta x_s, \tag{3}$$

where the second relation of Eq. (3) is obtained by taking Taylor expansion of the integrand to the first order and $\langle \partial_{x_s} h \rangle$ is averaged over the region between $x_s$ and $x_s + \delta x_s$. The third relation of Eq. (3) results from the onset condition for a local slip to occur[31,36], when the slope $k_0$ of the external elastic pulling force, $F_e(x_0; x_s) = k_0(x_0 - x_s)$, becomes equal to the local (downward) slope $-\langle \partial_{x_s} h \rangle L_y$ of the asperity-induced pinning force field,

$$F_i(x_s) = \int_{L_y} dy h(y; x_s). \tag{4}$$

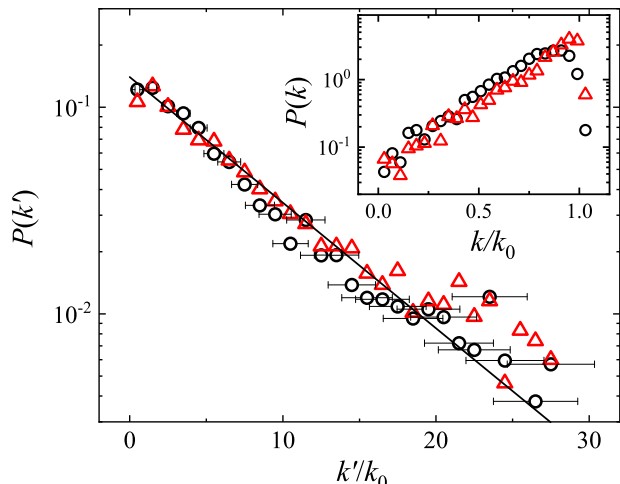

**Fig. 5 | Statistics of the local force gradient.** Measured PDF of the local force gradient $k'$ of the pinning force field. In the plot, the value of $k'$ is normalized by $k_0$. The measurements are made at the same scanning speed $U = 100$ nm/s, and under the same normal load $N = 500$ nN, for both the quasi-1D probe (black circles) and 2D probe (red triangles). The error bars show the standard deviation of the black circles. The solid line shows an exponential fit, $P(k'/k_0) = b \exp[-b(k'/k_0)]$, to all the data with $b = 0.14 \pm 0.02$. The inset shows the PDF $P(k)$ of the normalized dynamic spring constant $k/k_0$. Source data are provided as a Source Data file. Source Data.

Indeed, we find that this onset condition is satisfied during the slip (see SI Section III.A for more details). Equation (3), therefore, establishes a direct connection between the force release $\delta f$ and slip length $\delta x_s$.

The interface motion is stuck on the uphill of the pinning force field $F_i(x_s)$ when the local (upward) slope, $k' = dF_i(x_s)/dx_s > k_0$. In this case, the elastic pulling force $F_e(x_0; x_s)$ is balanced by the local pinning force $F_i(x_s)$, from which we obtain (see SI Section III.A for more details)

$$\frac{1}{k} = \frac{1}{k'} + \frac{1}{k_0}, \qquad (5)$$

where $k$ is the dynamic spring constant measured when the scanning probe undergoes a steady stick-slip motion, as shown in Fig. 1f. Equation (5) states that the interface is often partially pinned, and its local (microscopic) slip allows the scanning probe to feel a local force gradient $k'$. As the scanning probe sweeps over the sandpaper, it feels different values of $k'$ and the measured $k$ varies correspondingly.

The inset of Fig. 5 shows the measured PDF $P(k)$ of the normalized dynamic spring constant $k/k_0$. The two sets of data for the quasi-1D probe (black circles) and 2D probe (red triangles) have a similar shape. The measured $k/k_0$ varies in the range $0 \lesssim k/k_0 \lesssim 1$ and is peaked around $k/k_0 \simeq 0.97$. Equation (5) indicates that $k$ has an upper bound $k_0$ when the scanning probe is completely pinned ($k' \to \infty$, as shown in Fig. 4), which explains the variation range of the measured $k/k_0$. With Eq. (5), one can subtract out the contribution of $k_0$ from the measured $k$ and obtain the local force gradient $k'$. Figure 5 shows the obtained PDF $P(k')$ of the normalized $k'/k_0$. The measured PDFs $P(k')$ have a heavy-tailed distribution and can be well described by a simple exponential function, $P(k'/k_0) = b \exp[-b(k'/k_0)]$. Exponential-like PDFs, which fall off much slower than a Gaussian, were observed in various dynamically or spatially heterogenous systems[37,38]. From the fitting result, we find the average value $\langle k' \rangle / k_0 = 1/b \simeq 7.1$ (for black circles). With $k' \simeq (2\pi^2 E_b)/\lambda^2$ for a hypothetical sinusoidal pinning force field, the PT model[29–31] predicted that the transition to stick-slip motion occurs when $\langle k' \rangle / k_0 > 1$. Figure 5 thus reveals that this condition is satisfied for our system.

Under the mean-field approximation, the stick-slip motion of the scanning probe may be envisioned as a result of its center-of-mass (at

position $x_s$) moving in a random pinning force field $F_i(x_s)$ in Eq. (4) under the influence of $F_e(x_0; x_s)$, where $x_0 = Ut$. The equation of motion of $x_s(t)$ can be written as

$$m\frac{d^2 x_s}{dt^2} + \gamma\frac{dx_s}{dt} = k_0(Ut - x_s) - F_i(x_s), \qquad (6)$$

where $m$ and $\gamma$ are, respectively, the effective mass and damping coefficient of the scanning probe, and $F_i(x_s) > 0$ is a random pinning force field. Equation (6) is an extension of the PT model, which laid down the foundation for our understanding of stick-slip instabilities. In the original PT model[19,29–31], the external pinning (or frictional) force field $F_i(x_s)$ was assumed to be of a sinusoidal form for a single crystalline surface. For many rough surfaces of practical interest, however, $F_i(x_s) > 0$ is a random force field. Because disorder has many different forms, modeling a realistic random force field that can be compared directly with the experimental results remains a challenging task. Here we show that because the asperity-induced frictional force acting on a mesoscale scanning probe is a running average of the individual pinning forces resulting from a finite number of asperities at the interface, the resulting force field $F_i(x_s)$ is Brownian correlated[39], i.e., $\langle |F_i(x_s) - F_i(x_s')|^2 \rangle = 2D|x_s - x_s'|$, where $D$ is a measure of the fluctuation amplitude of $F_i(x_s)$ (see SI Section III.B for more details). The Brownian correlation is a general feature of the random pinning force field and has been used in the Alessandro–Beatrice–Bertotti–Montorsi (ABBM) model[4,15,40] to describe the avalanche dynamics of domain walls in soft magnets.

Equation (6) can be transformed into the time evolution of the slip velocity $v(t) \equiv dx_s/dt$ can be obtained by solving the dimensionless form of Eq. (6) (see SI Section III.B for more details)

$$\frac{d^2 V}{dt^2} + \gamma'\frac{dV}{dt} = 1 - V + \sqrt{V}\eta(t), \qquad (7)$$

where $V = v/U$, $\gamma' = \gamma/\sqrt{mk_0}$, $t$ has a unit of $\sqrt{m/k_0}$, and $\eta(t)$ is a Gaussian white noise with $\langle \eta(t)\eta(t') \rangle = 2D'\delta(t - t')$, and $D' = D/\sqrt{mk_0^3 U^2}$. For $D' > 1$, Eq. (7) gives rise to stick-slip motion. We numerically integrate Eq. (7) using the Euler–Maruyama method, following the protocol adopted in ref. 15. A slip starts at a zero acceleration ($\frac{dV}{dt}(t = 0) = 0$) and a very small positive speed $V(t = 0) \ll 1$ and stops when $V$ first becomes negative at $t = T_s$. The slip length $\delta x_s$ is then defined as $\delta x_s \equiv \int_0^{T_s} V dt$. With repeated slip events, we obtain the PDF $P(\delta x_s)$ in the parameter space $[\gamma', D']$.

In the under-damped regime ($\gamma' \leq 2$), we find $P(\delta x_s)$ obeys a power-law distribution, $P(\delta x_s) \sim \delta x_s^{-\epsilon}$, when $D' > 1$. An example is shown in the inset of Fig. 6 (see SI Section III.C and Section III.D for more numerical results). As shown in Fig. 6, the power-law exponent $\epsilon$ is well described by the relation, $\epsilon = \tau - \kappa/D'$ with $\kappa = 2.0 \pm 0.2$, and saturates at the asymptotic value, $\tau = 1.20 \pm 0.05$, when $D' \gg 1$. The error bar of the fitting parameters quoted here represents the confidence interval that we obtain from the fitting. Figure 6 also reveals that the value of $\gamma'$ has little effect on the power-law distribution of slip lengths so long as the system remains in the under-damped regime, i.e., when $\gamma' \leq 2$. A similar relation was also found in the ABBM model, where the power-law exponent $\epsilon$ is well described by the relation, $\epsilon = 3/2 - (1/2)/(D'\gamma')$[15] (see SI Section III.E for more details).

In the over-damped case, the power-law exponent $\epsilon$ can be analytically derived, and the second term is interpreted as a finite velocity correction. In the under-damped regime, the obtained asymptotic value of $\tau = 1.2$ is smaller than that predicted by the ABBM model ($\tau = 3/2$)[4,15] for the over-damped avalanche dynamics. This difference may be explained by the fact that the under-damped system has less dissipation, and more slip events with larger slip lengths are observed.

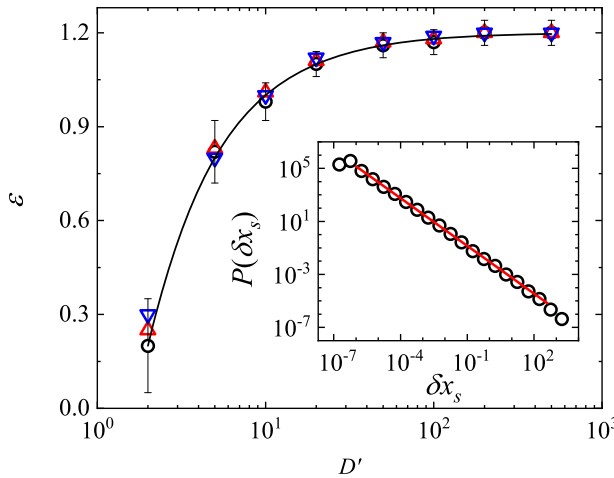

**Fig. 6 | Numerically calculated power-law exponent of the slip length distribution.** The data are obtained for three different values of $\gamma'$ in the underdamped regime: $\gamma' = 0.5$ (black circles), $\gamma' = 1$ (red upward triangles), and $\gamma' = 2$ (blue downward triangles). The solid line shows a fit, $\epsilon = \tau - \kappa/D'$, to the data points with $\tau = 1.20 \pm 0.05$ and $\kappa = 2.0 \pm 0.2$. Inset shows the obtained PDF $P(\delta x_s)$ of the slip length $\delta x_s$ from the numerical simulations of Eq. (7) at $D' = 500$ and $\gamma' = 1$ with a total of $1.5 \times 10^4$ slip events. The red solid line shows a power-law fit, $P(\delta x_s) \sim \delta x_s^{-\epsilon}$, to the data points with $\epsilon = 1.20 \pm 0.03$. The error bars show the standard deviation of the black circles. Source data are provided as a Source Data file. Source Data.

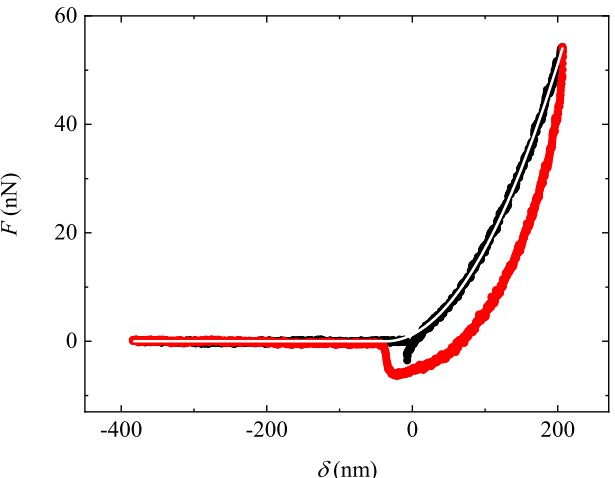

**Fig. 7 | Measurement of the Young's modulus of the UV-cured glue.** A typical force-indentation curve $F(\delta)$ is obtained when a conical AFM tip is pressed against a nanoparticles-embedded UV-cured glue surface with indentation depth $\delta$. The black and red curves are obtained, respectively, when the AFM tip advances and then retracts from the glue surface. The measurements are made at a constant speed of 0.64 μm/s. The white solid line shows a fit of Eq. (8) to the black curve with $E = 3.0$ MPa and $\delta_0 = -37$ nm. Source data are provided as a Source Data file.- Source Data.

The predicted value of $\tau = 1.2$ in the strong pinning regime (with $D' \gg 1$) is very close to the experimental result for the quasi-1D probe, and thus the model described by Eq. (6) provides a long-sought physical mechanism for the avalanche dynamics in stick-slip friction. Note that the relation, $\epsilon = \tau - \kappa/D'$, as shown in Fig. 6, is obtained only numerically, and we have not been able to derive the relation analytically from Eq. (6) up to now. Further theoretical study is needed.

Our results reveal that seemingly chaotic stick-slip friction at mesoscale obeys the statistical laws that are often associated with the avalanche dynamics at a critical state. These statistical laws remain invariant in the range of normal loads studied (see SI Section II.D for more details). In particular, we find that the measured PDFs of the slip length $\delta x_s$, the maximal force $F_c$ needed to trigger local slips, and the local force gradient $k'$ of the pinning force field are statistically interconnected. The broad distribution of $k'$ indicates that the stick-slip dynamics of the contact interface is caused primarily by fluctuations in the pinning force field. Slips with a larger slip length are lower-probability events, which require a larger value of $k'$ (or larger asperities) to hold the local contacts and a larger value of $F_c$ to de-pin them; both have a lower probability in their distributions. The proposed underdamped spring-block model under a Brownian-correlated pinning force field captures the essential physics of the stick-slip friction at mesoscale.

## Methods
### Assembly of the scanning probe
The assembly of the scanning probe is conducted under a high-magnification stereo-microscope using a micro-manipulator system. A rectangular silicon beam with a length $\ell \simeq 90\ \mu m$, width $w \simeq 35\ \mu m$ and thickness $t \simeq 2\ \mu m$ is glued to the front end of a commercial AFM cantilever (CSC37, MicroMasch) using a UV-curable glue (NOA 81, Norland). The hanging beam plane is made perpendicular to the cantilever plane, and its intersection with the cantilever orients parallel with the long axis of the cantilever, as shown in Fig. 1b, c. Once the beam is glued to the cantilever, the free end of the hanging beam is enclosed by a drop of the UV-curable glue mixed with 16 wt% silicon dioxide nanoparticles of mean size ~20 nm, which is then cured by exposure to a UV light. Finally, the glue part of the hanging beam is milled using a FIB (Helios G4 Dualbeam, Thermo Scientific) to generate

a flat rectangular end surface with the desired dimensions. The end surface of the hanging beam is made at an angle of 11° with respect to the cantilever surface so that it will remain horizontal (parallel to the substrate) when the probe is mounted to the AFM cantilever holder.

The Young's modulus $E$ of the UV-cured glue (together with the nanoparticles) is measured using a conical AFM tip. In the AFM measurement, a millimeter-sized drop of the UV-cured glue is deposited on a coverslip. By pressing the AFM tip against the glue surface, we obtain a force-indentation curve $F(\delta)$, as shown in Fig. 7. Because of the surface adhesion between the AFM tip and the UV-cured glue, the measured $F(\delta)$ in the advancing and receding directions does not overlap and show some hysteresis. The surface adhesion has less effect on the measured $F(\delta)$ in the advancing direction, which is well described by the Sneddon contact model[41],

$$F = \frac{\pi E}{2(1 - \nu^2)} \tan(\phi)(\delta - \delta_0)^2, \qquad (8)$$

where $\nu$ is the Poisson ratio of the glue, $\phi$ is the half-cone angle of the conical AFM tip, and $\delta_0$ is an offset of the indentation depth $\delta$. By taking $\nu = 0.5$ and $\phi = 20°$, we obtain a good fit to the data with $E = 3.0$ MPa. This value of $E$ is much smaller than Young's modulus of the sandpaper and the hanging beam (~170 GPa[42]). As a result, the glue-coated scanning probe provides better compliance at the contact when it is pushed against the sandpaper under a normal load. Meanwhile, mixing the nanoparticles into the glue reduces the adhesion of the probe to the sandpaper and strengthens the mechanical properties of the glue[43].

As shown by the SEM image in Fig. 1b, the end surface of the vertical silicon beam in the quasi-1D probe is covered by some uneven spotty surface texture, indicating that the end surface of the silicon beam is covered by a layer of the UV-cured glue and does not protrude through the glue. Because the end surface of the glue drop initially added to the free end of the vertical silicon beam has a limited size, it is quite difficult to cut a square end surface out of the glue drop alone with a size over 10 μm. To fabricate a 2D probe with dimensions of $12 \times 12\ \mu m^2$, we cut the glue drop slightly deeper so that a small portion of the vertical silicon beam is also cut through. As a result, both the

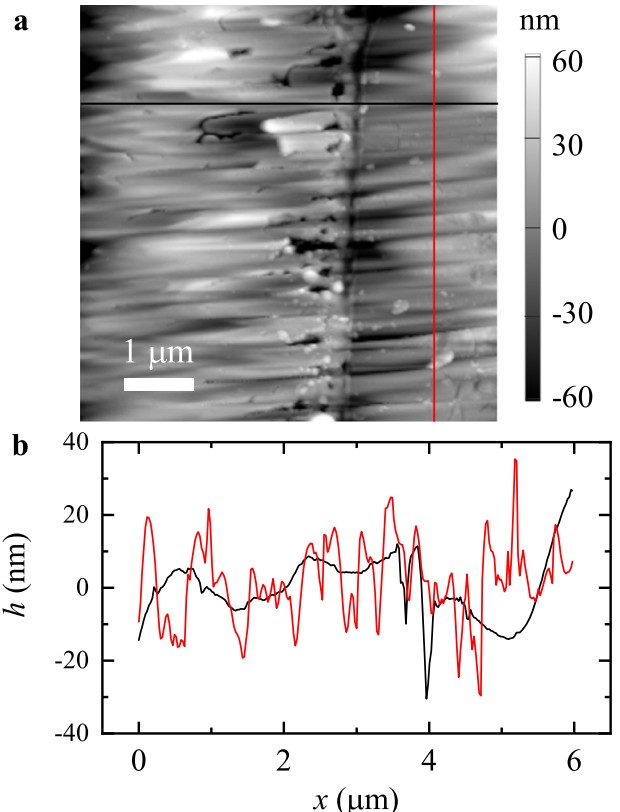

**Fig. 8 | Roughness of the end surface of the scanning probe. a** An AFM topographical image (6 μm × 6 μm) of the end surface of the 2D scanning probe, which is measured in the tapping mode. The vertical dark boundary between the left and right regions with parallel stripes indicates the position of the thin cantilever beam embedded in the middle of the end portion of the scanning probe. The vertical grayscale indicates the surface height. **b** Typical horizontal (black curve) and vertical (red curve) surface height profiles $h(x)$ along the black and red lines as marked in (**a**). Source data are provided as a Source Data file.Source Data.

vertical silicon beam and UV-cured glue remain on the same end surface, but the silicon beam itself does not stick out of the end surface. From the SEM image (end view) of the 2D probe shown in Fig. 1c, we find that the end surface of the vertical silicon beam has a sharp image without the uneven spotty surface texture associated with the glue layer, as shown in Fig. 1b. When the 2D probe is (spring) loaded onto the sandpaper, the end surfaces of the silicon beam and glue are both in close contact with the sandpaper surface. Because the contact area of the silicon beam ($2 \times 12$ μm$^2$) is small compared with the total contact area ($12 \times 12$ μm$^2$) of the 2D probe and the two contact regions as springs are connected in parallel, the silicon beam's contribution to the contact compliance, which is proportional to the contact area fraction, is also small (<17%). Therefore, the exposure of the vertical silicon beam in the contact area should not play an essential role in determining the contact mechanics of the 2D probe.

Figure 8a shows an AFM topographical image of the end surface of the 2D scanning probe used. The AFM image reveals that the end surface of the canning probe has some horizontal stripes with a typical width of sub-micrometers. These stripes are caused by the horizontal parallel milling with the FIB. Because the orientation of these parallel stripes is parallel to the scanning direction of the 2D probe (see the SEM image of the 2D probe in Fig. 1c), their influence on the stick-slip motion of the scanning probe (if any) will be minimal. As shown in Fig. 8b, the root-mean-squared (RMS) roughness of the end surface of the 2D probe is 20.5 nm over an area of 6 μm × 6 μm. This RMS roughness is about five times smaller than that of the sandpaper surface (99.7 nm over an area of 10 μm × 10 μm). Therefore, we conclude

that the asperities at the interface result mainly from the rough surface of the sandpaper, and the rough end surface of the scanning probe plays a less important role. In fact, the experimental findings reported above are the general features of stick-slip friction at the mesoscale, and they are valid even though some of the asperities at the interface come from the rough end surface of the scanning probes.

The entire hanging beam probe has flexibilities and, as a whole, acts as a spring so that the contact area between the end surface of the probe and the substrate is spring-loaded when a normal load $N$ is applied. In this case, any small angular mismatch $\theta$ between the two parallel contact surfaces can be treated as a weak spring connected in series with the probe, which can be self-aligned under the normal load $N$. This self-alignment effect can be estimated using the equation, $N = k_n(L/2)\tan\theta \simeq k_n(L/2)\theta$, where $k_n$ is the spring constant of the cantilever in the normal direction, and $L$ is the length of the contact area perpendicular to the scanning direction. Given that $N = 200–600$ nN, $k_n \simeq 0.7$ N/m, and $L \simeq 34$ μm, we have $\theta \simeq 2N/(k_nL) \simeq 1.0–2.9°$. With the focused ion beam, we were able to assemble the hanging beam probe with its end surface in parallel with the cantilever to within 1°. The above estimate thus suggests that the scanning probe can indeed maintain close contact with the substrate when the normal load $N$ is in the working range of this study.

Other details about the sample characterization and experimental methods are given in SI Section I.

## Data availability
Source data for the figures in the main text and supplementary information are provided in this paper. Raw data generated in this study are available from the corresponding author (P.T.) upon request. Source data are provided in this paper.

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

## Acknowledgements

This work was supported in part by RGC of Hong Kong under grant Nos. 16300920 (P.T.) and 16300421 (P.T.) and by MoST of Taiwan under the grant Nos. 110-2112-M-008-026-MY3 (P.Y.L.) and 110-2112-M-008-030- (H.Y.C.). P.Y.L. and H.Y.C. also acknowledge support by NCTS of Taiwan.

## Author contributions

C.Y. and P.T. designed the experiments; C.Y. performed the experiments; H.Y.C. and P.Y.L. developed the theoretical model with inputs from C.Y. and P.T.; all authors analyzed and interpreted data; C.Y. and P.T. drafted the paper with inputs from H.Y.C. and P.Y.L.; P.T. coordinated the project.

## Competing interests

The authors declare no competing interests.
