## [Peer Review File · Nature Communications]

Statistical laws of stick-slip friction at mesoscaleEditorial Note: This manuscript has been previously reviewed at another journal that is not operating a transparent peer review scheme. This document only contains reviewer comments and rebuttal letters for versions considered at *Nature Communications*.

REVIEWER COMMENTS

Reviewer #2 (Remarks to the Author):

Please see attachment. 
Reviewer #3 (Remarks to the Author):

Recommendation: Minor revisions.

The manuscript entitled “Statistical laws of stick-slip friction at mesoscale” by Yan et al. uses a hanging-beam lateral atomic force microscope to study stick-slip frictional force fluctuations at the mesoscale and to show that these stick-slip events follow avalanche dynamics.

The issues raised by the reviewers have been very well addressed and the manuscript has much improved now. Thus, I would recommend this manuscript for publication in *Nature Communications* once the following few points have been addressed:

1. The sentence “The free end of the hanging beam is attached with a drop of UV-cured glue...” could be misleading (page 2, 1st column). Better use: “The free end of the hanging beam is coated with a drop of UV-cured glue...”.
2. The term “height” or “height value” should be used for the scales in the captions of Figs. 1d, S1 and S3 instead of “height variations”. The AFM images show the variations, not the scales.
3. Please indicate the zero line for each of the three traces in Figs. 1 e,f. This is important, because they are offset.
4. What do the authors want to express with “bounded from above” (page 7, 1st column)?
5. The authors should use more cautious words when claiming that a certain value is small or even negligible, when it accounts to 17% (page 8, 2nd column) and almost 20 % (“...there is a pull-off force of 37 nN for both the quasi-1D and 2D probes. This adhesion force between the probe and sandpaper is small when compared with the normal load N used in the experiment, which is in the range of 200-600 nN.”, SI, page 6, 1st column).
6. Equations in the SI should be enumerated with an “S” to clearly distinguish them from the equations in the main text.
7. How does the E modulus value of 5 MPa for the UV glue (SI, page 3, 2nd column) result from the value of 3 MPa (main text, page 7, 2nd column)?
8. The pull-off forces mentioned by the authors are based on single force-extension curves (SI, page 6,

1st column and Fig. S10). However, statistics (mean and error values taken from a significant number of curves) would be required to properly compare those forces (i.e., are the differences really significant?).

9. "The slip length δx_s is the force release $\delta f/k_0$ normalized by the static spring constant k_0 ." (SI, page 8, 1st column) should read: "The slip length δx_s is the force release δf normalized by the static spring constant k_0 ."

10. "Equation (13) indicates that the measured k is an effective spring constant of two springs connected in series with one has a spring constant..." (SI, page 12, 2nd column) should read: "Equation (13) indicates that the measured k is an effective spring constant of two springs connected in series, wherein one has a spring constant..."

Reviewer #4 (Remarks to the Author):

Please see the PDF document attached. 
Response to Referee:

The authors wish to thank the referee for his/her careful review and constructive suggestions on this work. The following are the changes made in response to each of the referee's comments.

Referee #2 (Remarks to the Author):

The authors of "Statistical laws of stick-slip friction at mesoscale" do a comprehensive job of clearly addressing several of the concerns in the prior review. Nevertheless, there are some points of disagreement, and as such recommend further revision or rejection of the paper, with a preference for the former since the vast majority of the paper is both scientifically sound and an important contribution to the literature. The single remaining, specific concern is outlined below:

We appreciate that the authors identified our error in thinking the log-linear plot in original Fig S3a was a log-log plot. We made an additional error, however, which was implicitly equating the autocorrelation function, which the authors show, with the power spectral density, which the authors do not show. Our initial concern that the surface was fractal was its subjective appearance in Fig S1. To check, we calculated the power spectral density of the top half of the image in Fig S1a which results in the following:

This spectrum has typical features of a real fractal (within bounds) surface [1,2]: there is a low wavenumber cutoff ($\log k < 1.5$, due to the end of self-similarity which is why fractally rough samples can look flat at macroscale), and approximately linear fractal regions on a log-log plot. Here is a published example on ultrananocrystalline diamond [2], with the AFM measurement in blue:

(c) crude (e.g. converting grayscale to arbitrary height, only approximately correct lateral

image dimensions, subset of image analyzed, etc.), and the authors should produce and present an exact calculation in their response.

The authors' interpretation of the autocorrelation function is worth considering further. The authors say that the correlation length of an exponential decay of this function "represents a typical lateral dimension (or size) of the grain clusters (or asperities)". There are no citations to support this contention. We also conducted an abbreviated literature search to find similar use of the autocorrelation function in this context and could find no compelling examples, though of course this does not prove such examples don't exist, and their interpretation of the correlation length is intuitively satisfying. We tried this methodology with a synthetic image of monodisperse (diameter = 81mm) spheres on a flat surface, which is a reasonable proxy for a surface with a characteristic grain size. We find that, indeed, an exponential fit to the autocorrelation plot produces a correlation length of ~29mm, which is reasonably close to the sphere radius:

So it appears that the authors' approach can work if there truly is a characteristic grain size. Nevertheless, there seems to be clear evidence of a fractal surface roughness on the sandpaper over several decades of length, which aligns with an existing

literature about such surfaces and so we cannot, absent especially compelling justification, accept the correlation length as a proxy for some characteristic asperity size.

References

- [1] A. Gujrati, S. R. Khanal, L. Pastewka, T. D. B. Jacobs, *ACS Appl. Mater. Interfaces* **2018**, *10*, 29169.
- [2] A. Gujrati, A. Sanner, S. R. Khanal, N. Moldovan, H. Zeng, L. Pastewka, T. D. B. Jacobs, *Surf. Topogr.: Metrol. Prop.* **2021**, *9*, 014003.

We thank the referee for letting us know the recent work on surface characterization of various rough surfaces. Following the referee's suggestion, we calculated the power spectrum density (PSD) function $S(q)$ of surface height variations $h(x)$ of the sandpaper using the AFM topographical data. Because the sandpaper surface is isotropic, herein we only consider the one-dimensional PSD. The results are shown in the figure on the left. In the plot, the bottom axis is the wave vector $q = 2\pi/\delta$ in unit of m^{-1} and the top axis is the wave

length δ in μm . The units were chosen to compare with the results shown in the references given above. The red and black symbols show the obtained $S(q)$ from two AFM images: one is in Fig. 1(d) of the main text with a spatial resolution of $0.039 \mu\text{m}$ and the other is in Fig. S1(a) of the SI with a spatial resolution of $0.049 \mu\text{m}$. The two PSD functions obtained at two different spatial resolutions overlap well in the common range of the wave-vector q .

The measured $S(q)$ has a similar shape as that mentioned in the review report. Over the entire q -range, it can be described by two regimes of approximately power-law scaling, $S(q) \sim q^{-\alpha}$, together with a narrow intermediate cross-over region centered around $q \sim 10^7 \text{ m}^{-1}$. The power-law exponent α in the small- q regime (or long-wavelength regime with $\delta > 2 \mu\text{m}$) is $\alpha \simeq 1.6$ (left dashed line) whereas the value of α in the large- q regime (or short-wavelength regime with $\delta < 0.3 \mu\text{m}$) is $\alpha \simeq 3.5$ (right dashed line). While the cross-over region has a finite width, we find that the two power-law fits to the data meet at a cut-off length $\ell_c \simeq 0.34 \mu\text{m}$.

A power-law-like regime in the measured PSD suggests that the rough surface has some self-similar features and such a surface is referred to as a fractal. By examining the AFM images (Fig. S1(a) and Fig. 1(d) of the main text) of the sandpaper surface, we find that the power-law-like regime at small δ (from $\sim 0.1 \mu\text{m}$ to $\sim 0.4 \mu\text{m}$) can be attributed to the individual gains (with an average grain size $0.1 \mu\text{m}$) and their small clusters (such as dimers, trimers and tetramers). A similar conclusion was also drawn in Ref. 2 mentioned in the above review report. On the other hand, the power-law-like regime at large δ (from $\sim 2 \mu\text{m}$ to $\sim 30 \mu\text{m}$) is caused primarily by the random distributions of larger clusters (aggregation of small clusters), which are associated with

the bright regions (or bumps) and dark regions (or depressions) in the AFM topographical images. The cut-off length ℓ_c that separates the two power-law-like regimes, therefore, can be viewed as the lower cutoff of the large cluster (or domain) sizes. The above figure (new Fig. S2(b)) thus demonstrates that the sandpaper has various micro-contacts (or asperities) with a broad range of sizes and we, therefore, agree with the referee that the surface roughness of the sandpaper cannot be simply described by a single characteristic (asperity) size.

For our system, the maximum micro-contact size is limited by the dimension of the apparent contact area along the scanning direction, which is $\sim 4 \mu\text{m}$ for the quasi-1D probe and $\sim 12 \mu\text{m}$ for the 2D probe. The minimum micro-contact size, on the other hand, is the size of the grain particles ($0.1 \mu\text{m}$) on the sandpaper. Within the two length scales, the measured PSD (as a function of wavelength δ) has two (short-ranged) power-law-like regimes, which nonetheless cannot be described by a single fractal. Further study is needed in order to understand how the size distribution of the asperities (or microscopic contacts) affects the avalanche dynamics of local slips. Given that the main text of the present paper is already 9-page-long and the Supplemental Material is 19-page-long, we felt that while having such a study is important in its own right, it is nevertheless beyond the scope of the present study.

According to the Wiener-Khinchin theorem, the PSD $S(q)$ is the Fourier transform of the auto-correlation function $C(\lambda)$. For an exponential auto-correlation function $C(\lambda) = \sigma_h^2 \exp(-\lambda/\lambda_0)$, where σ_h is the RMS roughness, its Fourier transform has a Lorentzian form, $S(q) = (2/\pi)^{1/2} \sigma_h^2 \lambda_0 / (1 + (\lambda_0 q)^2)$. The blue solid line in the above figure ($S(q)$ vs. q) is a plot of the Lorentzian with $\sigma_h = 0.97 \mu\text{m}$ (the measured RMS roughness is $1.0 \mu\text{m}$) and $\lambda_0 = 0.48 \mu\text{m}$. It is seen that the Lorentzian PSD can only cover a fraction of q -range ($10^6 < q < 2.5 \times 10^7$) but not the entire range of the wave vector. Our previous fit of a simple exponential function to the measured $C(\lambda)$ was not perfect. The measured $C(\lambda)$ has a noisy tail and only the fast-decaying portion of $C(\lambda)$ has an exponential-like form. Incidentally, the method of auto-correlation function and PSD is well developed for the study of temporal signals, such as the laser light intensity $I(t)$ in dynamic light scattering (e.g., Cummins and Swinney, Light Beating Spectroscopy, in *Progress in Optics*, vol. VIII, p. 133, Ed. by E. Wolf, 1970) and has been widely used in the study of Brownian motion of colloidal particles (e.g., Clark, Lunacek, and Benedek, A Study of Brownian Motion using Light Scattering, *Am. J. Phys.* **38**, 575 (1970)) and critical fluctuations (e.g., Goldburg, The Dynamic of Phase Separation near the Critical Point: Spinodal Decomposition and Nucleation, in *Scattering Techniques Applied to Supramolecular and Non-equilibrium Systems*, Ed. by Chen, Chu and Nossal, 1981). For spatial signals, such as the surface height profile $h(x)$, one needs to exchange the variables $x \leftrightarrow t$ and $q \leftrightarrow \omega$.

To address the referee's concerns and make the above points clearer, we removed the discussion on the auto-correlation function $C(\lambda)$ and replaced Fig. S2 (b) by the above figure. In addition, we added the above discussions and references to the SI (p.1, Sec.

I.A, starting from the 3rd paragraph). We also added a paragraph in SI discussing the effect of fractal-like rough surfaces, as proposed by Filippov and Popov in Ref. 29 (p.16, left column, 2nd paragraph). In the main text, we changed the phrase “the typical size of grain clusters” to “the lower cutoff size of the grain clusters” (p.3, left column, 2nd paragraph).

Response to Referee:

The authors wish to thank the referee for his/her careful review and constructive suggestions on this work. The following are the changes made in response to each of the referee's comments.

Referee #3 (Remarks to the Author):

Recommendation: Minor revisions.

The manuscript entitled “Statistical laws of stick-slip friction at mesoscale” by Yan et al. uses a hanging-beam lateral atomic force microscope to study stick-slip frictional force fluctuations at the mesoscale and to show that these stick-slip events follow avalanche dynamics.

The issues raised by the reviewers have been very well addressed and the manuscript has much improved now. Thus, I would recommend this manuscript for publication in Nature Communications once the following few points have been addressed:

1. The sentence “The free end of the hanging beam is attached with a drop of UV-cured glue...” could be misleading (page 2, 1st column). Better use: “The free end of the hanging beam is coated with a drop of UV-cured glue...”.

The change is made as suggested (p.2, left column, 1st paragraph).

2. The term “height” or “height value” should be used for the scales in the captions of Figs. 1d, S1 and S3 instead of “height variations”. The AFM images show the variations, not the scales.

The phrase “height variations” is changed to “height” in the captions of Figs. 1d, S1 and S3.

3. Please indicate the zero line for each of the three traces in Figs. 1 e,f. This is important, because they are offset.

Figure 1(e) is too crowded to include additional zero lines. To address the referee's concern, we revised the figure caption to make it clear that the new zero-points of the vertical axis for the black and red curves are, respectively, 0.15 μN and 0.5 μN .

4. What do the authors want to express with “bounded from above” (page 7, 1st column)?

To address the referee's concern, we changed the sentence to “bounded by the physical dimensions of the apparent contact area” (this paragraph is now moved to SI on p.17, right column, last paragraph before Sec. III.D).

5. The authors should use more cautious words when claiming that a certain value is small or even negligible, when it accounts to 17% (page 8, 2nd column) and almost 20 % (“...there is a pull-off force of 37 nN for both the quasi-1D and 2D probes. This adhesion force between the probe and sandpaper is small when compared with the normal load N used in the experiment, which is in the range of 200-600 nN.”, SI, page 6, 1st column).

Following the referee’s suggestion, we changed the phrase “... is negligible under ambient conditions” to “... is small compared with the applied normal load under ambient conditions” (main text, p.2, right column). As mentioned in the answer to Question 8, we repeated the force measurements at 10 different locations on the same substrate and obtained the mean value and the standard deviation of the adhesion force to be 21 ± 5.8 nN for the quasi-1D probe over the sandpaper. This adhesion force is indeed small compared with the normal load N (200-600 nN) used in the experiment (SI, p.7, left column, 1st paragraph).

6. Equations in the SI should be enumerated with an “S” to clearly distinguish them from the equations in the main text.

All the equation numbers in the SI are changed as suggested.

7. How does the E modulus value of 5 MPa for the UV glue (SI, page 3, 2nd column) result from the value of 3 MPa (main text, page 7, 2nd column)?

We measured the Young’s modulus of the UV-glue again during the previous revision and updated the value in the main text but did not do so in SI. Now the Young’s modulus in SI is updated together with the shear modulus G and k_t^{max} (SI, p. 4, left column, 3rd paragraph).

8. The pull-off forces mentioned by the authors are based on single force-extension curves (SI, page 6, 1st column and Fig. S10). However, statistics (mean and error values taken from a significant number of curves) would be required to properly compare those forces (i.e., are the differences really significant?).

Following the referee’s suggestion, we repeated the force measurements at 10 different locations on the same substrate and obtained the mean value and the standard deviation of the adhesion force for different probes and substrates. In the revised Sec. II.B, we reported the new values of the mean and standard deviation of the adhesion force for different probes and substrates (SI, p. 7, left column, 1st paragraph). In the revised Fig. S10, we showed four typical contact-force-versus-distance curves (SI, p. 7, left column).

9. “The slip length δx_s is the force release $\delta f/k_0$ normalized by the static spring constant k_0 .” (SI, page 8, 1st column) should read: “The slip length δx_s is the force release δf normalized by the static spring constant k_0 .”

Following the referee's suggestion, the sentence is changed to "The slip length δx_s is the force release δf normalized by the static spring constant k_0 , i.e., $\delta x_s = \delta f/k_0$." (SI, p.8, left column, last paragraph).

10. "Equation (13) indicates that the measured k is an effective spring constant of two springs connected in series with one has a spring constant..." (SI, page 12, 2nd column) should read: "Equation (13) indicates that the measured k is an effective spring constant of two springs connected in series, wherein one has a spring constant..."

The correction is made as suggested (SI, p.14, right column, after Eq. (S13)).

Response to Referee:

The authors wish to thank the referee for his/her careful review and constructive suggestions on this work. The following are the changes made in response to each of the referee's comments.

Referee #4 (Remarks to the Author):

Yan and coworkers characterise the statistical behaviour of the stick-slip dynamics of a mesoscale contact. The authors present the results of a novel mesoscale friction experiments where a deformable AFM probe slides over a rough, hard surface. The systematic experimental data on the critical force, dynamic spring constant and slip-length are described in terms of different statistical distribution. The experimental statistical behaviour is rationalised in terms of numerical simulations based on a PT model modified to include a stochastic force field reminiscent of the ABBM model.

The work is novel and interesting, with possible implications in many fields, from the mesoscopic avalanche dynamics on which the authors focus to nanoscale friction and macro-scale earthquake dynamics. The experimental setup is benchmarked extensively and the results convincing. Moreover, the authors have improved considerably the manuscript following the first round of review. At the same time, we find that the theoretical part is still lacking and more in-depth analysis of the proposed model and a physical interpretation of the parameters are needed.

The work has the possibility of being of interested for a wide audience encompassing different communities (as required for a prestigious venue such as Nature Communication), but more effort is needed to explain the physics underpinning the model presented, to integrate the work in the literature and to outline the scope of validity of the results. Hence to warrant publication in Nature Communication the following major points need to be addressed in full.

Major points:

1. The authors claim that the dynamics of the system at $N=100$ nN is characterised by smooth sliding rather than stick-slip because of the absence of sawtooth-like features in the force signal (green curve in Fig. 1e,f). This is not a sufficient proof for the claim. The transition from smooth-sliding to stick-slip should be detected from the qualitative different dissipation behaviour in the two regimes, i.e. from the evolution of the dissipation as a function of sliding velocity or from the hysteresis in friction force loops [1]. Regarding the first method, the stick-slip dissipation increases logarithmically with velocity, while the dissipation in the smooth-sliding regime is linear with the velocity. Regarding the second method, forward-backward friction loop shows no hysteresis in the smooth sliding regime. The authors already report these types of analysis for loads

in the “intermediate range” (Fig. S8 and Fig. S17). To support the claim of smooth sliding, the same analysis must be repeated for the low load $N = 100\text{nN}$.

Following the referee’s suggestion, we conducted additional measurements of the frictional force in the small load regime. Figure (a) on the left shows the frictional force loops at a fixed scanning speed but under three different normal loads. At $N = 100$ nN, the force trajectory shows continuous fluctuations with a few abrupt changes (slips) occasionally. As N decreases, the number of small asperities decreases and so does the magnitude of the measured force fluctuations. The residual rare slip events become fewer and eventually disappear at $N = 10$ nN. The hysteresis between the mean frictional forces in the advance direction and in the receding direction also decreases

with decreasing N . Such a reduction of friction-induced hysteresis with decreasing normal load was also observed in atomic friction at the single contact limit (Ref. 16 in SI).

Figure (b) shows how the measured frictional force loops change with the scanning speed U when the normal load is held at a small constant value of $N = 20$ nN. It is seen that the measured frictional force loops do not reveal any significant speed-dependence in the small load regime. This result is in agreement with a general trend that was observed for rough surfaces (Refs. 19, 20) that the dynamic friction coefficient (a ratio of the frictional force to the normal load) in steady sliding is roughly a constant or decreases slightly with increasing U when U is small, with a nonzero quasi-static limit as $U \rightarrow 0$. This result, however, does not agree with those obtained in atomic friction at the single contact limit, in which the frictional force goes to zero as $U \rightarrow 0$, which is referred to in the above review report. Here the situation is changed when the apparent contact area contains a finite number of micro-contacts or asperities, which are randomly distributed both in space and in size. In this case, the effect of the energy barriers associated with different asperities cannot be averaged out coherently, as that in atomic friction at the single contact limit, and thus gives rise to a non-zero frictional force at the quasi-static limit as $U \rightarrow 0$.

Following the referee’s suggestion, we added the above figure (new Fig. S18) and a new discussion section (Sec. II.G) in the SI (p.11). The reference given in the review report and several other relevant references are cited in the discussion.

2. The authors highlight the fact that the exponent $\alpha = 1.2$ differs from the exponent 1.5 found in the overdamped limit in Ref. [2] (which is Ref [4] in the main text). In the same reference, Figure 10 shows, in the overdamped and quasi static limit, a power-law distribution for the avalanches size with exponent 1.5 for long-range elastic interactions and 1.27 (closer to the one identified by the authors in this system) for the short-length elastic interactions. Moreover, Fig. 11 in Ref. [2] reports the power law distribution when relaxing the quasi static assumption, i.e. the limit $c \rightarrow 0$ in the ABBM model where c is rate of change of the imposed field (which should relate to the tip speed U in the authors’ modified PT model). For $c > 0$ the exponent of the avalanche size decreases from 1.5 to 1.15. In Fig. S17b, what would be the exponent of the power law for δx_s for the increasing velocities reported? Can the authors justify their assumption of under-damped regime and comment on why their results cannot be explained in the overdamped limit considering a finite sliding velocity? Moreover, to establish a better comparison with the literature on the ABBM model and benchmark their own modified PT model, the authors should report the results of their model in the overdamped limit as well ($\gamma' > 2$ according to section III.B).

As mentioned in the main text (p. 6, after Eq. (6)), our model (i.e., Eq. (6)) is an extension of the Prandtl-Tomlinson (PT) model, which has been widely used in the study of dry friction and nanotribology. The PT model was considered as the most successful and influential “minimalistic” model in rationalizing the wealth of nanoscale and mesoscale friction data produced over the last decades (see the review articles of Ref. 19 and Ref. 32 in the main text). A number of experiments have been conducted and demonstrated the characteristic underdamped features of dry friction. For example, attenuated oscillations after a slip were observed in the friction-force traces obtained using a surface force apparatus with two mica surfaces sliding across a solidified molecular film (J. Klein, Phys. Rev. Lett. **98**, 056101 (2007)). Overshoots of a single slip over multiple slip-lengths were found in the lateral force trajectories obtained when a sharp AFM tip slides over a crystalline solid surface (S. N. Medyanik *et. al*, Phys. Rev. Lett., **97**, 136106 (2006)). These measurements clearly demonstrated that dry friction in the single-asperity limit is underdamped and thus the PT model is deemed appropriate to describe many fundamental properties of dry friction.

The present experiment broadens the study of dry friction into the regime of multi-asperity dynamics and so does our model, which provides an intermediate level description bridging the gap between the microscopic behavior of individual slips and the macroscopic laws of solid friction. Because the decay rate that determines how long the inertia effect will last scales as γ/m , which is independent of the system size (or mass), dry friction in the multi-asperity regime will remain underdamped as that in the

single-asperity regime. As a result, the predictions of the ABBM model and its variations as mentioned in the above referee report, which are valid only for the overdamped systems, will not be applicable to dry friction at mesoscale.

To make the above points clearer, we added two discussion paragraphs on the PT model in SI (p.15, right column, first two paragraphs).

The referee is quite correct in pointing out that our model will effectively reduce to the ABBM model when $m = 0$ in Eq. (S14) or $\gamma' \gg 1$ in Eq. (S21). In this case, Eq. (S21)

has only one dimensionless control parameter, namely, D'/γ' . Following the referee's suggestion, we conducted a numerical study of the avalanche dynamics in the over-damped case, using Eq. (S21) with a large value of $\gamma' = 100$. The numerical protocol used is the same as that for the under-damped case. Figure (a) on the left shows the obtained PDFs $P(\delta x_s)$ of the slip length δx_s in the over-damped regime (with $\gamma' = 100$) for different values of D'/γ' . It is seen that the obtained $P(\delta x_s)$ for different values of D'/γ' all follows the power-law distribution, $P(\delta x_s) \sim \delta x_s^{-\epsilon}$. Figure (b) on the left shows how the power-law exponent ϵ increases with D'/γ' over a wide range of D'/γ' . At the strong pinning limit ($D'/\gamma' = 10^2$), the obtained power-law exponent, $\epsilon \approx 1.5$, which is in excellent agreement with the prediction of the ABBM

model. Note that the strong pinning limit in Eq. (S21) requires $D'/\gamma' \gg 1$. It is seen that our numerical results agrees well with the prediction of the ABBM model, $\epsilon = \frac{3}{2} -$

$\frac{1/2}{D'/\gamma'}$ (solid line). Figure (b) thus demonstrates that our numerical protocol of simulating the slip events works well and that the avalanche dynamics in the under-damped regime is very different from that in the over-damped regime.

Following the referee's suggestion, we added the above figure (new Fig. S23) and a new discussion section (Sec. III.E) in the SI (p.19).

3. In deriving Eq. 17 in the SI, the authors assume that the “effective interfacial tension”

$g(x; x_s)$ has finite variance σ_g^2 . How does this assumption reflect on the shear field $\sigma(x, y; x_s)$, which is the real physical quantity at the base of their model? We understand that the model based on this assumption describes well the experimental results and, thus, this is an appropriate assumption for the system at hand. At the same time, given the universality claimed by the authors, it would be helpful to understand how this crucial assumption extended to other surfaces. For example, would this assumption still be valid for the rough surfaces characterised in Ref. [3,4] (also Ref. [1] in the SI), assuming that the power spectral density in Ref. [3,4] can be related to the surface topography, as it seems given that k' (Fig. S14) and $C(\lambda)$ (Fig. S2b) follow the same statistic?

As pointed out by the referee, the shear stress $\sigma(x, y; x_s)$ itself is a real physical quantity and thus has a finite variance. Similarly, the frictional force $F_i(x_s)$ is also a real physical quantity and thus has a finite variance as well. Because $F_i(x_s)$ is a two-dimensional integral over the contact area A , $F_i(x_s) = \iint_A [\sigma(x, y; x_s) dy] dx$, the integrand in the bracket, i.e., $g(x; x_s) = \int \sigma(x, y; x_s) dy$, will also have a finite value. The statement that “Assuming $g(x; x_s)$ has a zero mean and a variance σ_g^2 ” simply means that here we only consider the part of $\sigma(x, y; x_s)$ that contributes to the fluctuating part of the frictional force, $F_i(x_s) - F_0$, where $F_0 = \langle F_i(x_s) \rangle$ is the mean value of $F_i(x_s)$. In the revised SI (p. 15, left column, last paragraph), we have explained that the constant frictional force F_0 only gives rise to an additional steady-state stretching F_0/k_0 to the pulling spring without changing the fluctuation dynamics, as described in Eq. (S14).

In fact, the above statement is true even for fractal-like rough surfaces. This is because the self-similar feature of the actual solid surfaces always has two cutoff length. The lower cutoff is determined by the smallest asperity size (e.g., the grain size of the sandpaper), whereas the higher cutoff must be bounded by the system size. Consequently, the “effective interfacial tension” $g(x; x_s)$ will also have a finite variance σ_g^2 , which is related to the force variance σ_F^2 by $\sigma_F^2 = \sigma_g^2 L_x$, where L_x is the boundary length of the contact area in the x direction. For a multi-contact system, there is no meaning to calculate the frictional force at length scales below the smallest asperity size.

To make the above points clearer, we revised the discussion on the properties of $g(x; x_s)$ in the SI (p.16, left column, 2nd & 3rd paragraphs). In addition, we changed the symbol $g(x; x_s)$ to $h(x; x_s)$ to keep a consistent notation.

4. The author should compare their results to other models attempting to describe mesoscopic stick-slip dynamics. Filippov and Popov [5] considered a modified Prandtl-Tomlinson model with a random potential with a defined fractal dimension, instead of the standard sinusoidal potential. While in Ref. [5] the physics underpinning the

modified PT model is different and the analysis focuses on how the dissipation depends on the velocity, the mean-field model described in Eq. 3 therein and the one presented in Eq. 14 of the SI are closely related. Could the author comment on these similarities and whether they would expect the dynamics of a single asperity sliding on a fractal surface to obey the same statistical laws presented in their study for an ensemble of interacting asperities?

We thank the referee for letting us know this interesting reference. As mentioned by the referee, the paper by Filippov and Popov focused on how the dissipation depends on the scanning velocity and did not give any prediction on the probability distribution of slip length δx_s , which is the main focus of the present paper. In fact, we are not aware of any previous theoretical study of modified PT models, which gave specific predictions on the power-law distribution of slip length δx_s and its scaling exponent.

There are a number of experimental studies aimed at finding the scaling property of various rough surfaces (Refs. 1-3 in SI). Following the referee's suggestion, we calculated the power spectrum density (PSD) function $S(q)$ of roughness height variations $h(x)$ of the sandpaper using the two AFM images shown in Fig. 1(d) of the main text and in Fig. S1(a) of the SI. The results are shown in the revised Fig. S2 (b). It is found that the measured $S(q)$ for the sandpaper has a similar shape as those obtained in the previous studies (Refs. 1-3 in SI). Over the entire q -range ($10^5 \text{ m}^{-1} < q < 10^8 \text{ m}^{-1}$), the measured $S(q)$ can be described by two regimes of approximately power-law scaling, $S(q) \sim q^{-\alpha}$, together with a narrow intermediate cross-over region. The power-law exponent α in the small- q regime (or long-wavelength regime with $\delta > 2 \text{ }\mu\text{m}$) is $\alpha \simeq 1.6$ (left dashed line) whereas the value of α in the large- q regime (or short-wavelength regime with $\delta < 0.3 \text{ }\mu\text{m}$) is $\alpha \simeq 3.5$ (right dashed line). Figure S2(b) thus demonstrates that the sandpaper surface has various asperities (or micro-contacts) with a broad range of sizes. Nonetheless, the sandpaper surface cannot be described by a single fractal over the wavelength range that is relevant to our experiment.

At the moment, we are not aware of any theoretical model, which establishes a quantitative relation between the surface roughness and the actual pinning force field $F_i(x_s)$. This issue is further complicated by the size of the scanning probe, which feels the actual pinning potential. When a sharp AFM tip scans over a fractal-like rough surface, the tip can feel the entire roughness landscape of the underlying surface. In this case, the resulting slip lengths may follow a power-law distribution, which is linked to the statistics of the surface roughness or the pinning force field. Because it is solely determined by the self-similar nature of the surface roughness, the power-law distribution in this single-asperity regime does not have any connection with the avalanche dynamics. In this work, we demonstrate that the stick-slip friction in mesoscale is an emergent (or self-organized) phenomenon, which is realized in the

multiple-asperity regime. In this case, the power-law distribution of slip length δx_s arises from the spontaneous response of multiple asperities at the contact to the stick-slip instabilities and is not necessarily linked to the self-similar nature of the surface roughness. There is only a general requirement for the rough surface under study that the corresponding pinning force field $F_i(x)$ is random enough and Brownian-correlated. While the study of the modified Prandtl-Tomlinson model with a fractal-like pinning potential is certainly interesting in its own right, it is nevertheless beyond the scope of the present study.

Following the referee's suggestion, we included the results of the measured PSD function $S(q)$ in the new Fig. S2 (b). In addition, we added a full discussion on rough surface characterization and cited related references in the SI (p1, left column, Sec. I.A, starting from the 3rd paragraph). We also added a paragraph in SI discussing the effect of fractal-like rough surfaces, as proposed by Filippov and Popov (p.16, left column, 2nd paragraph). Their paper was cited as Ref. 29.

5. The procedure to study numerically the modified PT model is problematic. In the literature, the PT model is usually solved by integrating a Langevin equation (like Eq. 14 in the SI) and analysing the steady-state trajectory of the COM [5,6]. On the other hand, overdamped systems like the ABBM model are often treated with methods based the Fokker-Plank equation [2,7]. Instead the authors study the underdamped dynamics induced by Eq. 14 by integrating Eq. 21 with an ad hoc protocol (as far as we are aware) where the variable (the rescaled velocity $V = vU$) evolves from the same initial condition ($V(T = 0) = 10^{-4}$) until it first become negative; the integration is then restarted with a different seed for the white noise ξ and the same initial condition. What is the physical meaning of the first term in Eq. 21, i.e. the derivative of the rescaled acceleration? As no references for this protocol are provided, the authors must show that this protocol is equivalent to a standard integration of the Langevin equation Eq. 14 or provide a solution based on the Fokker-Plank equation. It would be instructive to show the stick-slip motion of the COM sliding over the random pinning field for at least one significant case, e.g., $D'=500$. From such a steady-state trajectory one could directly compute the distribution of the slip-length and compare it with the results reported in Fig. S19.

The numerical method adopted in this study has been used in a previously study (see Ref. [15] in the main text) for not only the avalanche size distribution but also for other avalanche properties, such as the averaged avalanche shape and maximum velocity distribution. This method has an advantage that no threshold value is needed to identify the slip events from a time series of the steady-state velocity (see more discussions below). As shown in the answer to Question #2, we used the same method to study the avalanche dynamics of the overdamped case and found an excellent agreement with the

analytical results. We therefore do not find any problem associated with this numerical method. In SI, we have shown that Eq. (S21) is a direct mathematical transformation of Eq. (14), whose physical meaning has been clearly explained. Therefore, the physical origin of each terms in Eq. (S21) should be traced back to those in Eq. (S14).

To make these points clearer and to address the referee’s concern, we added Ref. [15] in the introduction of the numerical protocol in the main text (p.6, right column, 1st paragraph).

Figure (a) on the right shows an example of a steady-state trajectory $V(T)$ obtained by solving Eq. (6) in the main text with a realization of the Brownian-correlated random pinning force field. The dimensionless parameters used here are $\gamma' = 2$ and $D' = 500$. It is

seen that $V(T)$ exhibits alternating quiet periods (stick state) and burst periods (slip state), indicating the stick-slip dynamics. Figure 2(b) shows a magnified view of $V(T)$ during the period of $T = 165 - 200$. It is seen that the large velocity burst, which is marked by two red crosses to indicate its start and end times, is followed by a decaying oscillation. This oscillation is caused by the underdamped nature of Eq. (6). Because the pinning field is random, the shape of the oscillation is irregular and cannot be modelled by a well-defined function, such as that for the simple damped oscillation in the absence of the random pinning force field. It is thus very difficult to set up a uniform threshold value to distinguish the “after-slip” oscillations from the actual small slip events in the time series of the steady-state velocity. As mentioned above, our numerical protocol effectively avoids this problem.

To make the above points clearer, we added the figure (new Fig. S22) and a discussion section (Sec. III.D) in SI (end of p.17).

6. Another problem linked to this protocol is the underdamped nature of the model. By stopping and restarting the simulation as soon as the slip velocity becomes negative, the system loses any memory of the previous slips: each new slip is oblivious of the residual kinetic energy accumulated during the previous ones. This means that the dynamical effects that would occur in an underdamped dynamics are effectively removed, i.e. the system is actually overdamped. This conceptual problem makes it crucial to compare the numerical results obtain with this protocol with a steady-state trajectory obtained from Eq. 14.

As shown in the figure in the answer to Question #5 (new Fig. S22), the underdamped dynamics as described by Eq. (6) (or Eq. S14) generates both large burst (slip) events and small “after-slip” oscillations. It is seen from the figure (Fig. S22) that in the strong pinning or low speed limit (i.e., at the large value of $D' = 500$), the stick period between the neighboring large bursts (slips) is sufficiently long such that the system has enough time for the “after-slip” oscillations to die out. In this case, removing the small “after-slip” oscillations with our numerical protocol does not affect the statistical properties of the large burst (slip) events. It allows us to filter out unwanted noise and focus on the main effect of the large slip events, which are generated by the underdamped dynamics as well.

To make the above points clearer and to address the referee’s concern, we added the figure in the answer to Question #5 (new Fig. S22) and a discussion section (Sec. III.D) in SI (end of p.17).

7. The data points relative to $\gamma' = 0$ in Fig. S20 is puzzling. If no energy is removed from the system, how can it reach a steady state? This is another reason to compare these data with a corresponding trajectory of the COM.

Besides γ' , the random pinning force $F_i(x_s)$ (> 0) itself produces dissipation over a long time so that the system can reach a steady state. In fact, some theoretical models (such as that by J. M. Carlson and J. S. Lange, Phys. Rev. A, **40**, 11 (1989)) only used the frictional force term in the equation of motion and ignored the viscous damping term when describing the relative motion between two solid surfaces. Nevertheless, we agree with the referee that $\gamma' = 0$ is a special point and one needs to treat this condition with care.

To address the referee’s concern and to make the above points clearer, we removed the data obtained at $\gamma'=0$ and added a new set of data obtained at $\gamma'=0.5$ in Fig. 6 in the main text (the original Fig. S20 in the SI was removed because it repeated Fig. 6). In addition, we revised the discussion on Eq. (S14) in the SI (p.15, left column, last paragraph). To highlight the fact that the pinning force $F_i(x_s)$ is actually the frictional force, we changed the sign of the pinning force term from $+F_i(x_s)$ to $-F_i(x_s)$ and added the condition $F_i(x_s) > 0$ both in Eq. (6) of the main text and Eq. (S14) in SI. We deleted the condition $\langle F_i(x_s) \rangle = 0$ in Eq. (6) of the main text and Eq. (S14) in the SI, which is the convention used in the ABBM model and in some extensions of the PT model.

8. The authors should report the protocol used to fit the exponent in Fig. S20. As the distributions for $D' < 20$ do not show a clear power-law distribution, the protocol adopted is not self-evident. Moreover, what is the meaning of the parameter κ ?

The power-law exponent ε in Fig. S20 (now Fig. S22) is determined from a compensated plot of $P(\delta x_s)\delta x_s^\varepsilon$ as a function of δx_s . The value of ε is adjusted so that it gives the largest plateau region, in which we have $P(\delta x_s)\delta x_s^\varepsilon = \text{constant}$. The error bar of the final value of ε is determined by the fitting uncertainty of ε . The two figures below show the compensated plots of $P(\delta x_s)\delta x_s^\varepsilon$ at the two limit values of D' : $D' = 500$ (left) and $D' = 2$ (right). The former has the longest power-law region (about seven decades) and the latter has the shortest power-law region (about one decade). This method of using compensated plots to determine the power-law exponent is widely used in the study of the scaling properties of turbulent flows (see, e.g., Ahlers, Grossmann, and Lohse, Heat transfer and large-scale dynamics in turbulent Rayleigh-Benard convection, Rev. Mod. Phys. **81**, 503 (2009)).

Following the referee's suggestion, we revised Fig. S19 (now Fig. S21), in which the colored curves are shifted vertically so that the power-law fits and the power-law regions are clearly visible. We also added a discussion paragraph in Sec. III.C of SI, which describes the protocol used in the power-law fit, as discussed above (p.17, left column, last paragraph).

As shown in Fig. 6 of the main text, the fitted power-law exponent ε at different values of D' is well described by the relation, $\varepsilon = 1.2 - \kappa/D'$ with $\kappa = 2.0$. A similar relation was also found in the ABBM model, where the power-law exponent ε is well described by the relation, $\varepsilon = 3/2 - \kappa/(D'\gamma')$ with $\kappa = 1/2$. In the over-damped case, the power-law exponent ε can be analytically derived and the second term is interpreted as a finite velocity correction. Here $1/(D'\gamma') = k_0\gamma U/D$ and κ is just a numerical factor. Similarly, in the under-damped case, we have $1/D' = m^{1/2}k_0^{3/2}U/D$ and κ is another numerical factor. Therefore, the second term in the under-damped case can also be interpreted as a finite velocity correction. In the latter case, however, the relation $\varepsilon = 1.2 - \kappa/D'$ is obtained only numerically, and we have not been able to derive the relation analytically from Eq. (6) up to now. Further theoretical study is needed.

To make the above points clearer, we revised the discussion on this relation in the main text (p.6, right column, last paragraph; p.7, left column, 1st paragraph).

9. Fig. S19 reports the slip length distribution computed as the integral of the slip velocity V during the slip time T_s . While these two quantities cannot be obtained experimentally, they are available from the mean-field model: a more detailed analysis could help clarify the mechanism underpinning the observed distributions. Is T_s power-law distributed while V is constant, i.e. the slips have no characteristic time but they all happen at a well-defined speed? Or the opposite, i.e. the slips always take the same amount of time but they proceed at widely different speeds? Or, finally, the power law distribution of the slip-length δx_s arises from a complex interplay of the two underlying distributions?

Following the referee's suggestion, we examined the velocity time series $V(T)$ during the slips. The figure on the left shows four examples of $V(T)$ obtained at $\gamma' = 2$ and $D'=500$. It is seen that these slip events have different slip durations T_s and different maximum velocities. The velocity trajectories show an overall acceleration phase followed by a deceleration phase, but the actual shape of $V(T)$ varies for different slips, which is indeed a manifestation of the stochastic nature of avalanches. From the above figure (and the figure below), we find that neither the velocity V nor the slip duration T_s remain constant during the slips.

In addition, we calculated the PDF of the slip duration T_s for different values of D' and at a fixed value of $\gamma' = 2$. It is seen from the figure on the left that the slip duration T_s varies over a much narrower range compared with that of the slip length δx_s , which makes it difficult to tell if there is indeed a power-law regime for T_s . In the numerical study of the PDF of the slip length δx_s , we used at least two full decades of δx_s to confirm a power-law distribution. At the strong pinning limit ($D' = 500$), we find that the obtained $P(T_s)$ has a power-law region of approximately one and a half decades and the corresponding power-law exponent is

close to 1.9, which is slightly smaller than that predicted by the ABBM model ($= 2$). At the moment, we are not sure if the slip duration T_s indeed has a power-law distribution and if so whether the power-law exponent is very different from that for the over-damped case.

Clearly, a further study of the statistical properties of the slip velocity V and slip duration T_s is needed. As the main focus of this paper is on the statistical properties of the slip length, which are directly measured in the experiment, we felt that while having a complementary study of the slip velocity V and slip duration T_s is important in its own right, it is nevertheless beyond the scope of the present study.

To address the referee’s concern and to make the above points clearer, we added Fig. S20 and a discussion paragraph in SI (p.17, left column, 2nd paragraph).

10. There is an inconsistency between the assumption of underdamped regime and the way the authors estimate the slip-length δx_s from the force drop δf . The slip condition in Eq. 11 in the SI (or Eq. 3 in the main text) linking the slip length to the force drop is only valid in the overdamped regime, see Ref [8]. In an underdamped system the tip can overshoot the overdamped δx_s due to the undissipated kinetic energy. This mechanism could also explain the discrepancy between the onset condition and the steady-state condition reported below Eq. 13 in the SI. The author claim that a deviation of 10% is negligible without commenting on why or reporting what would be a non-negligible deviation. Moreover, we argue that this reported deviation might be the lowest in their dataset: the 10% deviation refers to the data at load $N = 400$ nN and $\langle k'/k_0 \rangle \approx 5.7$ (1D probe). The same analysis must be reported also for other experimental conditions, e.g. higher loads ($N = 500$ — 600 nN) and 2D probe with $\langle k'/k_0 \rangle \approx 7.1$, where the deviation could be higher according to the derivation in Section III.A in the SI.

To answer this question more clearly, we revised Fig. S19 as shown on the left. At the onset of slip, the CoM is at x_B . The end position of the CoM after the slip depends on the system damping. For an over-damped system, the CoM will jump to x_C and stay there. For a zero-damping system, the CoM will jump to x_D and then oscillates between x_D and x_C , such that the grey shaded area under the black dashed line is equal to the blue shaded area above the black dashed line. For a system with a moderate damping, the CoM will jump to a position between x_D and x_C , say x_E , and then undergoes a damped oscillation

and gradually drifts toward x_c , until the remaining kinetic energy is fully dissipated. Because of friction at the solid interface, the actual stick-slip motion is always influenced by some level of damping (see the answer to Question #7). As shown in the Table below, the use of the relation $\delta x_s \approx \delta f/k_0$ to estimate the slip length, which is valid for over-damped systems, is indeed a good approximation for our system with an accuracy better than 90%.

Probe	$\Delta x_s^{(b)}$	$N = 300$ nN	$N = 400$ nN	$N = 500$ nN	$N = 600$ nN
		$\Delta x_s^{(a)} (\frac{\Delta x_s}{\Delta x_0})$	$\Delta x_s^{(a)} (\frac{\Delta x_s}{\Delta x_0})$	$\Delta x_s^{(a)} (\frac{\Delta x_s}{\Delta x_0})$	$\Delta x_s^{(a)} (\frac{\Delta x_s}{\Delta x_0})$
quasi-1D	74.6 μm	386.5 μm (92%)	386.4 μm (92%)	379.4 μm (91%)	
2D	61.7 μm		405.4 μm (93%)	430.9 μm (99%)	442.2 μm (101%)

Following the referee's suggestion, we calculated the total displacements of the CoM in the stick phase, $\Delta x_s^{(b)} \approx \frac{\Delta x_0}{1+(k'/k_0)}$, and in the slip phase $\Delta x_s^{(a)}$ under different normal loads N . The final results are summarized in the Table above. It is seen that the total displacement $\Delta x_s = \Delta x_s^{(a)} + \Delta x_s^{(b)}$ of the CoM obtained under different experimental conditions are all very close to the scanning distance Δx_0 (with over 91% agreement). This result thus confirms that the relation $\delta x_s \approx \delta f/k_0$, which is used to calculate Δx_s , gives a good estimate of the slip length from the measured force drop. The small difference between Δx_s and Δx_0 ($< 9\%$) may be attributed to a small underestimation made by using the relation $\delta x_s \approx \delta f/k_0$, as discussed above, and/or the nonlinear effect of very large slip events. It is seen from the above Table that the statement in the above review report that "Moreover, we argue that this reported deviation might be the lowest in their dataset" is not true.

To address the referee's concern, we revised Fig. S19 and its discussions in SI (p.14, left column, last paragraph). We also added the above Table in SI and revised the discussion on the slip condition (p.14, right column, last paragraph).

Minor points:

A. In order to extended the findings of this work to other systems of interest for the wide readership of Nature Communications it would be helpful to understand the necessary conditions underpinning these results. For example, the interface is composed of two elements of very different hardness (Young modulus of the probe ~ 3 MPa and of the sandpaper ~ 170 GPa), how do the results depend on this large difference in hardness? Would the conclusion drawn here apply to other system where a multi-asperity contact is present, e.g. a colloidal probe sliding of graphite [9]?

The main purpose of using two contact materials with a large difference in hardness is to increase the compliance of the contact so that the number of asperities (or micro-contacts) with different random configurations (e.g., different roughness heights and

sizes) is maximized at the contact without introducing large plastic deformations and wear. More specifically, the rigid sandpaper surface used in the experiment serves as a base to provide a large number of asperities with a broad range of randomness, whereas the probe surface is made of a softer material and is flat so that it is able to have a close contact with the rough surface of the sandpaper at all length scales and feel the underlying pinning force field, when a small normal load is applied. If the probe surface were made of another rigid sandpaper, for example, the contact between the two hard surfaces is dominated primarily by a small number of the most prominent asperities on each surface. In this case, it would be difficult to have an effective local contact under a regular normal load that can feel many asperities with a broad range of randomness. In fact, Ref. [9] mentioned in the review report discussed such a case. When the colloidal probe (made of a rigid glass sphere) slides over a graphite surface, the glass sphere is coated by a layer of graphite flakes and the resulting contact is dominated by single-asperity stick-slip, even when the normal load is increased significantly in the experiment.

To make the above points clearer, we added a discussion paragraph in the SI (p.2, right column, 2nd paragraph).

B. The authors report all the results in the main text both for a 1D- and a 2D-probe. The two datasets follow the same statistical laws and differ quantitatively in the exponent of the slip-length distribution. Is there a physical meaning associated with this quantitative difference, e.g. the fact that with an exponent $\alpha = 1.2$ (1D probe) the median of the distribution is defined while for $\alpha = 0.7$ (2D probe) it is not? The only time this difference is discussed in the main text is to justify the fact that the modified PT model explains quantitatively the exponent of the 1D probe but not that of the 2D probe. This paragraph weakens the universality claimed by the authors (and a quantitative difference could be expected given the mean-field nature of the model), while the results for the 2D-probe in general strengthen it, showing that the behaviour does not depend on the specific geometry. Hence we suggest that the authors consider moving the discussion of the 2D probe to the SI, as its presence in the main text makes the discussion heavier and less clear without adding any qualitative difference compared to the 1D probe.

Following the referee's suggestion, we moved the discussion paragraph to the SI (p.17, right column, 2nd paragraph). As mentioned in the paragraph, the slip length δx_s measured in the experiment is actually bounded by the physical dimensions of the apparent contact area. Therefore, there is no divergent issue for the median of the distribution associated with the 2D probe. Whether a 2D pinning force field $F_i(x, y)$ to be included in Eq. (S14) or the dimensionality of the scanning probe can give rise to a

different universality class is an important open question, which needs further theoretical investigation. Nonetheless, it is beyond the scope of the present study.

C. Does the exponent $\alpha = 1.2$ for the slip length reported in the main text correspond to the exponent $\tau = 3/2$ for the avalanche size in Ref [2] (i.e. Ref. [4] in the main text)? In the same reference the exponent $\alpha = 2$ is associated to the avalanches duration. If this is the case, the authors should conform to the established naming convention or specify the difference in order to avoid confusion.

Following the referee's suggestion, we used the letter τ to denote the power-law exponent for the slip length (main text and SI).

D. In Section III.A of the SI (page 12) where the authors state "By taking a derivative of the stick-state condition [...]" would it perhaps be more correct to refer to it as a differential rather than a derivative, as the following paragraph deals with dx_s and dx_0 ?

The change is made as suggested (SI, p.14, right column, 2nd paragraph).

E. In the same paragraph (before eq. 13), it seems that k' and k are defined as the same quantity. Should the definition of the dynamic spring constant be $k = dF_i(x_s)/dx_0$ rather than $k = dF_i(x_s)/dx_s$, as the latter would be the same as k' ?

This is a typo. We made the correction $k = dF_i(x_s)/dx_0$ in the revised SI (p.14, right column, 2nd paragraph, above Eq. S13).

F. The units on the x axis in Fig. S2b should be " μm " and not "um".

The change is made as suggested (SI, p.2, Fig. S2b).

G. The scaling of $C(\lambda)$ is difficult to see in the main plot of Fig. S2b, we suggest reporting the data in log-log scale, as done in the answer to previous rounds of review. Moreover, the authors should consider showing the power spectral density as in Ref [3,4]: some readers could be more familiar with this quantity and reporting it would ease comparisons to the surface topography of this study. This should only amount to compute the FT of the height autocorrelation function already reported in Fig. S2b, as stated in Ref. [4].

Following the referee's suggestion, we calculated the power spectrum density (PSD) function $S(q)$ of surface height variations $h(x)$ of the sandpaper using the AFM topographical data. Because the sandpaper surface is isotropic, herein we only consider the one-dimensional PSD. The results are shown in the new Fig. S2(b). In the log-log

plot, the bottom axis is the wave vector $q = 2\pi/\delta$ in unit of m^{-1} and the top axis is the wave length δ in μm . The units were chosen to compare with the results given in the references mentioned in the above review report.

We thank the referee for letting us know the recent work on surface characterization of various rough surfaces. These references are cited in the revised SI (Refs. 1-3). Because the measured $S(q)$ already covered all the information about the surface roughness, we removed the discussion on the auto-correlation function $C(\lambda)$ and replaced Fig. S2 (b) with the new results for $S(q)$. In addition, we added three discussion paragraphs on $S(q)$ in the SI (p.1, Sec. I.A, starting from the 3rd paragraph).

H. While the claim that no plastic deformation of the cantilever takes place in the high-load limit is supported, the extension of the Greenwood and Williamson contact model at high-load proposed in Fig. S15 lacks a quantitative argument. The claim “Figure S15 thus shows an extension of the Greenwood and Williamson model for interacting asperities in the high-load regime.” is unsupported by any experimental evidence or theoretical analysis of this interaction. Another possible explanation of the different behaviour at high loads could be that the same asperities present at smaller loads undergo multiple slips rather than single ones, at such high loads [8]. While the behaviour at high loads does not influence the main message of the paper, the authors should make their claims quantitative or restructure this section into a qualitative discussion of the different statistical behaviour observed.

Following the referee’s suggestion, we revised the discussion on Fig. S15. In particular, we clarified that Figure S15 only provides an argument, which extends the Greenwood and Williamson model for interacting asperities in the high-load regime. In addition, we stated that both the effects of asperity coalescence and multi-asperity slips, where the entire scanning probe crosses over a number of asperities in a single slip event, are possible reasons for the observed stick-slip behavior in the high-load regime (SI, p.10, left column, 1st and 2nd paragraphs).

I. We suggest to the authors to include in the main text the sentence reported in answer to Reviewer 3 “The power-law exponent quantifies the relative incidence between the large-sized and small-sized slips, i.e., more large-sized slips are detected for a smaller value of α , and vice versa.”. This would clarify the physical meaning and importance of a power-law distributions to less-specialised readers.

The sentence is added in the main text as suggested (p.4, right column, end of the last paragraph).

J. We cannot find any content for Ref. [40] in the main text. Given that this is the only reference associated with the claim “Here we show that because the asperity-induced

frictional force acting on a mesoscale scanning probe is a running average of the individual pinning forces resulting from a finite number of asperities at the interface, the resulting force field $F_t(x_s)$ is Brownian correlated”, the author should provide an accessible reference to it.

Following the referee’s suggestion, we added new Ref. [40] (O. D. Anderson, Moving average processes, J. R. Stat. Soc. Series D (The Statistician), 24, 4 (1975)) in the revised main text. The original mathematical theorem is expressed in term of the autocorrelation function of a running average (or sum) of independent identical random variables (see Eq. (S16) in SI). In SI Sec. III.B, we showed that Eq. (S16) can be used to demonstrate that the mean-squared-difference function is Brownian correlated (see Eq. (S17) in SI).

K. Eq. 21 in the SI is a second order differential equation, and yet only on initial condition is reported, namely $V(T = 0) = 10^{-4}$. What is the other initial condition?

A slip starts at a zero acceleration and a small initial velocity $V_0 \equiv V(T = 0)$. The initial velocity V_0 only influences the lower cut-off of the power-law regime of the slip size distribution. The figure on the left shows the obtained $P(\delta x_s)$ from the simulations at fixed values of $\gamma' = 2$ and $D' = 500$ but with different V_0 . It is seen that a smaller value of V_0 gives rise to a smaller value of the lower cut-off. Here the obtained $P(\delta x_s)$'s are shifted vertically for a clear view. We chose $V_0 = 10^{-4}$ in this work in order to have a large power-law range.

To address the referee’s concern and to make the above points clearer, we added the condition that a slip starts at a zero acceleration and a small initial velocity in the numerical method discussed in the main text (p.6, right column, 1st paragraph). More details are given in the revised SI (p.16, right column, last paragraph).

To address the referee’s concern and to make the above points clearer, we added the condition that a slip starts at a zero acceleration and a small initial velocity in the numerical method discussed in the main text (p.6, right column, 1st paragraph). More details are given in the revised SI (p.16, right column, last paragraph).

References

- [1] A. Socoliuc, R. Bennewitz, E. Gnecco, E. Meyer (2004). “Transition from Stick-Slip to Continuous Sliding in Atomic Friction: Entering a New Regime of Ultralow Friction”. *Physical Review Letters*, 92(13), 134301.
- [2] F. Colaiori (2008) “Exactly solvable model of avalanches dynamics for Barkhausen crackling noise”, *Advances in Physics*, 57:4, 287-359

- [3] A. Gujrati, S. Khanal, L. Pastewka, T.D.B Jacobs (2018). “Combining TEM, AFM, and profilometry for quantitative topography characterization across all scales.” ACS Applied Materials & Interfaces.
- [4] T.D.B. Jacobs, T. Junge, L. Pastewka (2017). “Quantitative characterization of surface topography using spectral analysis”. Surface Topography: Metrology and Properties, 5(1), 013001.
- [5] A.E. Filippov, and V.L. Popov. “Fractal Tomlinson model for mesoscopic friction: From microscopic velocity-dependent damping to macroscopic Coulomb friction.” Physical Review E 75.2 (2007): 027103.
- [6] A Vanossi and O M Braun (2007) “Driven dynamics of simplified tribological models” J. Phys.: Condens. Matter 19 305017
- [7] P.C. Torche, T. Polcar, and O. Hovorka. "Thermodynamic aspects of nanoscale friction." Physical Review B 100.12 (2019): 125431.
- [8] S.N. Medyanik, W.K. Liu, I.H. Sung, R.W. Carpick (2006). “Predictions and observations of multiple slip modes in atomic-scale friction” Physical review letters, 97(13), 136106.
- [9] R Buzio, A Gerbi, C Bernini, L Repetto, A Vanossi (2022) “Sliding friction and superlubricity of colloidal AFM probes coated by tribo-induced graphitic transfer layers” Langmuir 38, 41, 12570–1258

We thank the referee for providing these useful references. Refs. [1,2,6,8] were cited in the revised main text, and Refs. [1-5,8] were cited in the revised SI. Refs. [7,9] are not directly related to the current work and thus they are not cited in this work.

REVIEWER COMMENTS

Reviewer #2 (Remarks to the Author):

The authors have addressed the concerns raised to my satisfaction.

Reviewer #3 (Remarks to the Author):

The manuscript entitled “Statistical laws of stick-slip friction at mesoscale” by Yan et al. has even further improved now, in particular due to the further explanations in the supplementary materials materials/ information. Thus, I would recommend this manuscript for publication in Nature Communications after the following few points are addressed:

1. The E-modulus of the UV curable glue is given as 3.0 MPa and the effective lateral spring constant is estimated based on that value (SI, page 4). At the same time, the authors claim: “Meanwhile, mixing the nanoparticles into the glue reduces the adhesion of the probe to the sandpaper and strengthens the mechanical properties of the glue [44].” (page 7, 2nd column, end of the column). If the mechanical properties are strengthened, wouldn't that mean that the E-modulus of the glue would then be much higher than 3.0 MPa affecting the effective lateral spring constant estimation?

2. The authors should pay attention to the number of digits:

Caption Fig. 1: (3 ± 0.5) μm should read (3.0 ± 0.5) μm .

Further examples:

a) Page 4, 2nd column, 3rd paragraph; caption Fig. 4; SI page 8, 2nd column, 1st paragraph; caption Fig. S13, SI page 9, 1st column, 1st paragraph; caption Fig. S17: $\tau = 1.12\pm 0.1$ and $\tau = 0.72 \pm 0.1$

b) Caption Fig. 6 and page 6, 2nd column, 2nd paragraph: $\tau = 1.2 \pm 0.05$ and $\epsilon = 1.2 \pm 0.03$.

c) Page 8, 2nd column, 3rd paragraph: $0.96\text{--}2.88^\circ$. Here the accuracy should be less when taking the accuracy of the initial values into account.

d) SI, page 7, 1st column: 21 ± 5.8 nN, 23 ± 7.3 nN, 59 ± 3.8 nN and 125 ± 2.4 nN

3. The supplementary materials/ information reads like a publication on its own rather than as a support of the claims of the main text. That could be improved by better referencing all sections of the supplementary materials/ information in the main text and possibly referencing the respective supplementary figures and tables.

4. Caption of Fig. 8 and of Fig. S2: “surface height variations” should be replaced by “surface height”.

5. SI, page 10, 1st column, 3rd paragraph: “spring contact of the cantilever” should read “spring constant of the cantilever”.

Reviewer #4 (Remarks to the Author):

The authors have implemented extensive revisions of the manuscript, supported by additional experiments and simulations. All the points we raised have been fully address and unclear passages have been clarified. In particular, the authors have proven the soundness of their numerical approach and correct the discussion on the theoretical model, integrating it better with existing literature on ABBM and PT models.

Hence, I am pleased to recommend this work for publication on my part, leaving to Referee #2 (who is clearly an expert on the topic) to assess the soundness of the authors' response on the implication of the now-clear fractal nature of the surface.

Response to Referee:

The authors wish to thank the referee for his/her careful review and constructive suggestions on this work. The following are the changes made in response to each of the referee's comments (marked in red).

Referee #3 (Remarks to the Author):

The manuscript entitled “Statistical laws of stick-slip friction at mesoscale” by Yan et al. has even further improved now, in particular due to the further explanations in the supplementary materials/ information. Thus, I would recommend this manuscript for publication in Nature Communications after the following few points are addressed:

1. The E-modulus of the UV curable glue is given as 3.0 MPa and the effective lateral spring constant is estimated based on that value (SI, page 4). At the same time, the authors claim: “Meanwhile, mixing the nanoparticles into the glue reduces the adhesion of the probe to the sandpaper and strengthens the mechanical properties of the glue [44].” (page 7, 2nd column, end of the column). If the mechanical properties are strengthened, wouldn't that mean that the E-modulus of the glue would then be much higher than 3.0 MPa affecting the effective lateral spring constant estimation?

The reported modulus value, $E \approx 3.0$ MPa, was actually measured for the nanoparticle-embedded glue, not for the pure glue. We mentioned this fact in the main text (p.7, right column, 1st paragraph): “The Young's modulus E of the UV cured glue (**together with the nanoparticles**) is measured ...”

To make this point clearer, we revised the sentences referring to the modulus of the nanoparticle-embedded glue in the main text (p.2, right column and p.7, caption of Fig.7) and SI (p4, left column, 3rd paragraph).

2. The authors should pay attention to the number of digits: Caption Fig. 1: (3 ± 0.5) μm should read (3.0 ± 0.5) μm . Further examples:

(a) Page 4, 2nd column, 3rd paragraph; caption Fig. 4; SI page 8, 2nd column, 1st paragraph; caption Fig. S13, SI page 9, 1st column, 1st paragraph; caption Fig. S17: $\tau = 1.12\pm 0.1$ and $\tau = 0.72 \pm 0.1$.

(b) Caption Fig. 6 and page 6, 2nd column, 2nd paragraph: $\tau = 1.2 \pm 0.05$ and $\epsilon = 1.2 \pm 0.03$.

(c) Page 8, 2nd column, 3rd paragraph: $0.96\text{--}2.88^\circ$. Here the accuracy should be less when taking the accuracy of the initial values into account.

(d) SI, page 7, 1st column: 21 ± 5.8 nN, 23 ± 7.3 nN, 59 ± 3.8 nN and 125 ± 2.4 nN.

Following the referee's suggestion, we changed the number of digits as follows:

Fig. 1 caption: (3.0 ± 0.5) μm ;

(a) Page 4, right column, 2nd paragraph; caption of Fig. 4; SI page 8, right column, 1st paragraph; caption of Fig. S13, SI page 9, left column, 1st paragraph; caption of Fig. S17: $\tau = 1.12\pm 0.10$ and $\tau = 0.72 \pm 0.10$;

(b) Caption of Fig. 6 and page 6, right column, 2nd paragraph: $\tau = 1.20 \pm 0.05$ and $\epsilon = 1.20 \pm$

0.03;

(c) The number of digits is reduced to two: 1.0–2.9°.

(d) SI, page 7, left column, 1st paragraph: 21 ± 6 nN, 23 ± 7 nN, 59 ± 4 nN and 125 ± 2 nN.

3. The supplementary materials/ information reads like a publication on its own rather than as a support of the claims of the main text. That could be improved by better referencing all sections of the supplementary materials/ information in the main text and possibly referencing the respective supplementary figures and tables.

Following the referee's suggestion, all the sections of the Supplementary Materials are now cited in the main text.

4. Caption of Fig. 8 and of Fig. S2: “surface height variations” should be replaced by “surface height”.

Corrections were made as suggested (main text, p.8, caption of Fig. 8 and SI, p.2, caption of Fig. S2).

5. SI, page 10, 1st column, 3rd paragraph: “spring contact of the cantilever” should read “spring constant of the cantilever”.

The typo was corrected (SI, p. 10, left column, 3rd paragraph).

REVIEWERS' COMMENTS

Reviewer #3 (Remarks to the Author):

The authors have addressed all remaining issues to my satisfaction. Thus, I recommend this work for publication!